# Activation of the integrated stress response by inhibitors of its kinases

**Maria Szaruga[1], Dino A. Janssen [ID][1,2], Claudia de Miguel[1,2], George Hodgson [ID][1,2], Agnieszka Fatalska[1], Aleksandra P. Pitera[1], Antonina Andreeva [ID][1] & Anne Bertolotti [ID][1] ✉**

Phosphorylation of the translation initiation factor eIF2α to initiate the integrated stress response (ISR) is a vital signalling event. Protein kinases activating the ISR, including PERK and GCN2, have attracted considerable attention for drug development. Here we find that the widely used ATP-competitive inhibitors of PERK, GSK2656157, GSK2606414 and AMG44, inhibit PERK in the nanomolar range, but surprisingly activate the ISR via GCN2 at micromolar concentrations. Similarly, a PKR inhibitor, C16, also activates GCN2. Conversely, GCN2 inhibitor A92 silences its target but induces the ISR via PERK. These findings are pivotal for understanding ISR biology and its therapeutic manipulations because most preclinical studies used these inhibitors at micromolar concentrations. Reconstitution of ISR activation with recombinant proteins demonstrates that PERK and PKR inhibitors directly activate dimeric GCN2, following a Gaussian activation-inhibition curve, with activation driven by allosterically increasing GCN2 affinity for ATP. The tyrosine kinase inhibitors Neratinib and Dovitinib also activate GCN2 by increasing affinity of GCN2 for ATP. Thus, the mechanism uncovered here might be broadly relevant to ATP-competitive inhibitors and perhaps to other kinases.

The ISR is a central controller of cell and organismal fitness elicited by phosphorylation of eIF2α on serine 51, which results in attenuating bulk protein synthesis as well as reprograming gene expression to adapt to and survive challenges[1,2]. In humans, this pathway is fine-tuned by the antagonistic action of four eIF2α kinases, GCN2, HRI, PKR and PERK, that sense different stimuli and two phosphatases, PPP1R15A-PP1 and PPP1R15B-PP1[3,4]. With its central role in cell survival, the ISR has emerged as an important therapeutic target with a broad range of possible applications[5,6]. A number of ISR kinase inhibitors have been developed[7,8]. PERK inhibition has been the focus of several large drug discovery programs, initially for oncology[7], that yielded multiple ATP-competitive inhibitors, GSK2656157 (GSK'157), GSK2606414 (GSK'414) and AMG44, with nanomolar affinities for their target[9–12]. Since ATP-competitive kinase inhibitors are prone to off-target effects, the PERK inhibitors were counter-screened against a panel of kinases and found to exert low-affinity inhibitory activity towards the other ISR kinases[9–11]. Over the past 10 years, these PERK inhibitors have been broadly used in 200 research papers (https://pubmed.ncbi.nlm.nih.gov/?term=%28%28GSK2606414%29+OR+%28GSK2656157%29%29+OR+%28AMG44%29) to dissect the role of the ISR in cells and organisms as well as to evaluate the therapeutic potential of ISR inhibition. Efficacy of ISR inhibitors was reported in a range of disease models, from cancer to neurodegeneration[7,8]. The majority of previous preclinical studies were conducted at doses above 25 mg/kg in vivo[7–9,12–15]. The resulting concentrations of the inhibitors measured in tissues were above 10 μM[9,12–15], a large excess for a nanomolar-affinity inhibitor. Such a dosing regime was selected to ensure saturated binding of PERK, a standard paradigm for kinase inhibition.

[1]MRC Laboratory of Molecular Biology, Francis Crick Avenue, Cambridge CB2 0QH, UK. [2]These authors contributed equally: Dino A. Janssen, Claudia de Miguel, George Hodgson. ✉e-mail: aberto@mrc-lmb.cam.ac.uk

Here, we report an unexpected dose-dependent property of ISR kinase inhibitors that has direct relevance for their use as tool compounds in basic research as well as their application in preclinical and clinical studies. We find that diverse ATP-competitive inhibitors of PERK inhibit it in the nanomolar range as expected[9–12], but surprisingly activate the ISR via GCN2 at micromolar concentrations. GCN2 activation is also observed with a PKR inhibitor. Conversely, GCN2 inhibitor A92 inhibits its target but induces the ISR via PERK. The GCN2-activating property of diverse ISR-kinase inhibitors is direct and follows a Gaussian activation-inhibition curve in cells as well as in vitro with recombinant proteins. We reveal the underlying mechanism showing that kinase activation by these compounds is driven by increasing GCN2 affinity for ATP. We also show that tyrosine kinase inhibitors Neratinib and Dovitinib, which were also found to activate GCN2[30], do so by increasing the kinase affinity for ATP. These findings have broad relevance for our understanding of the ISR in physiology and diseases as well as for the use and understanding of the mechanism of action of ATP-competitive kinase inhibitors.

## Results

### GSK'157 inhibits PERK but activates GCN2 and ISR in a dose-dependent manner

Tunicamycin (Tm) induces protein misfolding stress in the endoplasmic reticulum (ER), activating PERK in cells and in vivo[3,16]. Interested in modelling severe ER stress in vivo, we treated mice with Tm in the presence of a widely used ATP-competitive PERK inhibitor, GSK'157[11]. As expected, Tm caused PERK activation in mouse liver, resulting in its phosphorylation manifested by a mobility shift on SDS-PAGE, and increased phosphorylation of its substrate eIF2α in absence of GSK'157 (Fig. 1a). Surprisingly, whilst PERK inhibition was robust in vivo after administration of 50 mg/kg of the inhibitor, as previously reported[12], eIF2α phosphorylation was not inhibited (Fig. 1a). We therefore investigated how phosphorylation of eIF2α occurs in the presence of its inhibited kinase and turned to cells to elucidate the underlying mechanism.

In cells, Tm activated PERK and this was prevented by treatment with 0.02–10 μM of GSK'157 (Fig. 1b), as expected[12]. However, whilst phosphorylation of eIF2α and expression of the ISR marker ATF4 were inhibited at nanomolar concentrations of the inhibitor, GSK'157 increased these ISR signals from above 1 μM (Fig. 1b). Attenuation of bulk protein synthesis, a sensitive output of ISR induction[17], was evident 2 h after Tm treatment, followed by the expected recovery[17] (Fig. 1c). Whilst nanomolar concentrations of GSK'157 prevented translation attenuation upon Tm treatment due to PERK inhibition, as expected[11,12] (Fig. 1d), translation decreased in a dose-dependent and persistent manner at micromolar concentrations of the inhibitor (Fig. 1d, e). Thus, PERK inhibitor GSK'157 inhibits PERK and the ISR in the nanomolar range as anticipated[11,12] but surprisingly activates the ISR in the low micromolar range in ER stressed cells.

Searching for the molecular basis of this phenomenon, we reasoned that PERK inhibition might overwhelm another eIF2α kinase. Accumulation of misfolded proteins during unresolved ER stress could prevent their degradation and thereby diminish the pool of amino acids, a signal to activate GCN2, the amino-acid sensing ISR kinase[1,18]. We found that, in Tm-stressed cells, ISR-activating concentrations of GSK'157 caused robust activation of GCN2 detected by its phosphorylation and reduced mobility on SDS-PAGE (Fig. 1f). This demonstrates that, in the presence of ER stress, the benchmark PERK inhibitor GSK'157 activates GCN2 and the ISR in the low micromolar range.

### Activation of GCN2 by GSK'157 is entirely mediated by GCN2 independently of ER stress or PERK

We then examined whether the ISR-activating property of GSK'157 was dependent on ER stress. We found that GSK'157 activated GCN2, increased eIF2α phosphorylation and ATF4 expression, as well as

decreased protein synthesis in the absence of ER stress (Fig. 2a, b and Supplementary Fig. 1). This suggested that GSK'157 activates GCN2 independently of PERK. Indeed, GSK'157 activated GCN2 and induced a functional ISR in cells after PERK had been knocked down (Fig. 2c). This demonstrates that the ISR-activating property of GSK'157 is independent of its known target PERK. To test the possible involvement of other ISR kinases, we knocked-down each ISR kinase in HeLa cells (Fig. 2d and Supplementary Fig. 2). Inactivation of GCN2 but not HRI, PERK or PKR abolished the increase of eIF2α phosphorylation and ATF4 induction by GSK'157 (Fig. 2d). Likewise, decrease in translation upon GSK'157 was severely compromised upon GCN2 knockdown but occurred to control levels following knockdown of HRI, PERK or PKR (Fig. 2d). This demonstrates that ISR activation by GSK'157 is entirely mediated by GCN2. Thus, micromolar concentrations of the PERK inhibitor GSK'157 specifically activate GCN2 and induce a functional ISR in cells independently of ER stress or PERK.

### ISR kinase inhibitors activate the ISR

We then investigated if activation of GCN2 and the ISR by GSK'157 was limited to this specific compound. We found that the related PERK inhibitor GSK'414[10] also activated GCN2 and increased phosphorylation of eIF2α with similar potency to GSK'157 (Fig. 3a and Fig. 2a). As these compounds belong to the same chemical series, we next evaluated the activity of AMG44, a PERK inhibitor of a different chemotype[9]. AMG44 also induced GCN2 activation, resulting in eIF2α phosphorylation (Fig. 3d). The ISR activation observed following GSK'414 or AMG44 treatment was functional and resulted in attenuation of protein translation (Fig. 3b, c and Fig. 3e, f). Thus, PERK inhibitors of different chemotypes can activate GCN2 and the ISR.

Next, we asked if GCN2 inhibitors could also activate the ISR. Phosphorylated GCN2 was readily detectable in cell culture in absence of specific treatment, indicating basal activity (Fig. 3g), as expected[17]. Concentrations of the A92[19] GCN2 inhibitor above 0.63 μM inhibited GCN2 and decreased basal eIF2α phosphorylation (Fig. 3g). However, a dose-dependent increase in eIF2α phosphorylation was observed from 10 to 40 μM of A92 (Fig. 3g) and this was associated with activation of PERK kinase (Fig. 3g). Measurement of protein synthesis reflected these findings (Fig. 3h). Translation increased up to ~50% between 0.63 and 5 μM of A92 (Fig. 3h), coinciding with decreased phosphorylation of eIF2α (Fig. 3g). In contrast, high concentrations of the inhibitor that caused PERK activation and increased phosphorylation of eIF2α resulted in a decrease in protein translation (Fig. 3h, i). Knockdown of PERK, but not GCN2, abolished ISR activation by A92, establishing that its ISR-activating property is entirely mediated by PERK and independent of its primary target GCN2 (Supplementary Fig. 3). Thus, PERK and GCN2 inhibitors inhibit their primary targets as anticipated, but at high concentrations concomitantly activate the ISR via a sister eIF2α kinase.

### Reconstitution of GCN2 and ISR activation by PERK inhibitors in vitro

To elucidate the molecular mechanisms by which GSK'157 activates GCN2 and the ISR, we tried to recapitulate this activity with purified GCN2 kinase and its substrate eIF2α. For this, we used a previously described, active GCN2 fragment lacking the tRNA and ribosome binding regions[20,21] but containing its kinase and pseudokinase domains[22]. The GCN2 fragment was dimeric (Supplementary Fig. 4a, b) and was activated by addition of ATP in a concentration-dependent manner, which resulted in phosphorylation of eIF2α (Fig. 4a). GCN2 activation and eIF2α phosphorylation were not observed in control conditions lacking the kinase or ATP (Supplementary Fig. 5a). We next used a concentration of ATP that yielded limited activation of GCN2 (Fig. 4a) to test the putative potentiating activity of various compounds. GSK'157 activated GCN2 and increased eIF2α phosphorylation in a dose-dependent manner with maximal activation observed at ~5 μM

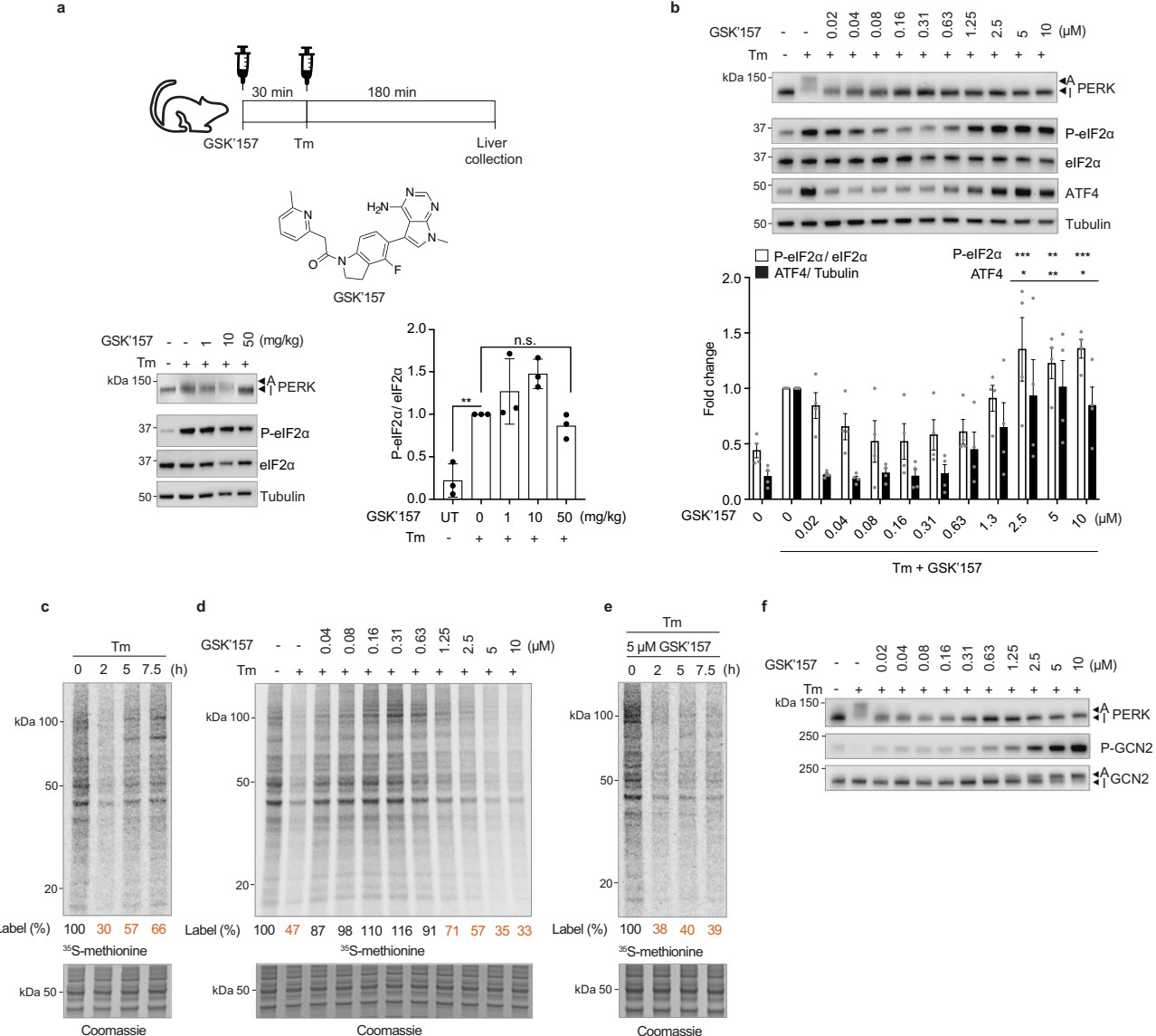

**Fig. 1 | Micromolar concentrations of PERK inhibitor GSK'157 activate GCN2 and the ISR in stressed cells. a** Schematic of mice treatments with 0.1 mg/kg Tm and increasing concentrations of GSK'157. Immunoblots of indicated proteins from lysates of liver samples from mice subjected to the indicated treatments. Active (A) and Inactive (I) PERK are indicated with arrows. Quantification of P-eIF2α and eIF2α from immunoblots such as the ones shown in the left panel. Data are shown as mean ± SD ($n = 3$), number of animals. **$p = 0.0046$; n.s.: not significant, as determined by one-way ANOVA with Dunnett's multiple comparison test. **b** Immunoblots of indicated proteins from lysates of HeLa cells pre-treated with GSK'157 for 30 min and then co-treated with Tm and GSK'157 for 2 h. Representative experiment from $n = 4$, biologically independent experiments. Ratio of P-eIF2α to eIF2α and ATF4 to Tubulin quantified from immunoblots and normalised to untreated cells. Data are shown as mean ± SEM ($n = 4$), biologically independent experiments. *$p ≤ 0.0412$, **$p ≤ 0.0067$, ***$p = 0.0006$, as determined by one-way

ANOVA with Dunnett's multiple comparison test. **c** Newly synthesized proteins labelled for 10 min with ³⁵S-methionine in HeLa cells following treatment with Tm for the indicated times and a Coomassie-stained gel as control. Representative experiment from $n = 2$, biologically independent experiments. **d** Same as panel (**c**), with HeLa cells pre-treated with GSK'157 for 30 min and then co-treated with Tm and GSK'157 for 2 h. Representative experiment from $n = 2$, biologically independent experiments. **e** Same as panel (**c**), with HeLa cells pre-treated with GSK'157 for 30 min and then co-treated with Tm and 5 µM of GSK'157 for the indicated times. Representative experiment from $n = 2$, biologically independent experiments. **f** Immunoblots of indicated proteins from lysates of HeLa cells pre-treated with GSK'157 for 30 min and then co-treated with Tm and GSK'157 for 2 h. Active (A) and Inactive (I) PERK and GCN2 are indicated with arrows. Representative experiment from $n = 4$, biologically independent experiments. Source data are provided as a Source data file.

(Fig. 4b, c). The in vitro activation of GCN2 by GSK'157 did not occur in absence of the kinase, or ATP (Supplementary Fig. 5a). Furthermore, activation of GCN2 by GSK'157 was inhibited by the ATP-competitive GCN2 inhibitor A92 (Fig. 4b, c). The in vitro results recapitulated the in vivo and cellular findings, and demonstrate that PERK inhibitor GSK'157 can directly activate GCN2 to phosphorylate eIF2α. After maximal activation, increasing concentrations of the compound progressively decreased activity of GCN2 and phosphorylation of its substrate, giving rise to a Gaussian activation-inhibition curve (Fig. 4b, c).

Having observed a Gaussian activation-inhibition pattern in vitro, we went back to cells to perform a large dose response of GSK'157. The cellular assays replicated the Gaussian activation-inhibition curve observed in vitro: GSK'157 activated GCN2 and induced the ISR between 1.25 µM to 20 µM whilst higher concentrations of the compound caused kinase inhibition and, as a result, the ISR was no longer activated (Supplementary Fig. 6a, b).

GSK'157 is an ATP-competitive inhibitor with an IC50 of 0.9 nM that inhibits PERK by occupying its ATP-binding site[11] (Fig. 4d).

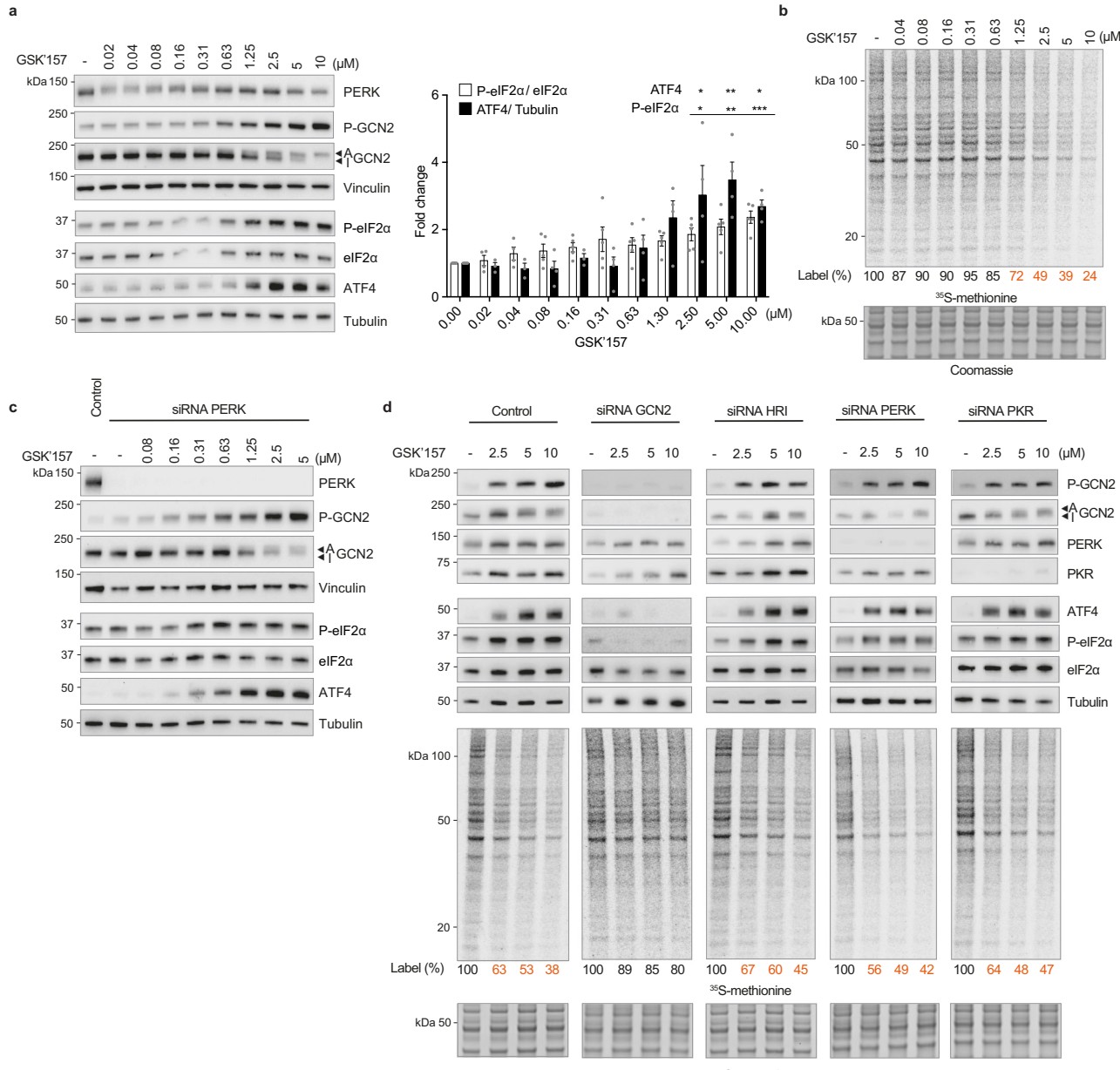

**Fig. 2 | GSK'157 activates GCN2 and the ISR independently of ER stress and PERK. a** Immunoblots of indicated proteins from lysates of HeLa cells treated with indicated concentrations of GSK'157 for 2.5 h. Active (A) and Inactive (I) GCN2 are indicated with arrows. Representative experiment from $n = 4$, biologically independent experiments. Ratio of P-eIF2α to eIF2α and ATF4 to Tubulin quantified from immunoblots and normalised to untreated cells. Data are shown as mean ± SEM (P-eIF2α $n = 5$ and ATF4 $n = 4$, except 0.02 and 0.04 μM compound treatment $n = 4$ and $n = 3$ respectively), biologically independent experiments. *$p \leq 0.0435$, **$p \leq 0.0038$, ***$p = 0.0002$, as determined by one-way ANOVA with Dunnett's multiple comparison test. **b** Newly synthesized proteins labelled for 10 min with 35S-methionine in HeLa cells treated with indicated concentrations of GSK'157 for 2.5 h and a Coomassie-stained gel as control. Representative experiment from

$n = 2$, biologically independent experiments. **c** Immunoblots of indicated proteins in lysates of HeLa cells untreated or pre-treated with PERK siRNA for 60 h and subjected to increasing concentrations of GSK'157 for 2.5 h. Active (A) and Inactive (I) GCN2 are indicated with arrows. Representative experiment from $n = 3$, biologically independent experiments. **d** Immunoblots of indicated proteins from lysates of HeLa cells untreated or pre-treated with indicated siRNA for 60 h and subjected to increasing concentrations of GSK'157 for 2.5 h. Active (A) and Inactive (I) GCN2 are indicated with arrows. Newly synthesized proteins in the same lysates labelled with 35S-methionine for 10 min and a Coomassie-stained gel as control. Representative experiment from $n = 2$, biologically independent experiments. Source data are provided as a Source data file.

The compound had been counter screened against other kinases and found to have low inhibitory activity towards GCN2 (IC50 of 3 μM)[11,12]. The rationale for this is the high homology between the ATP-binding sites[11,12]. The structure of the PERK kinase domain with GSK'157 has been reported[11] (Fig. 4d and Supplementary Fig. 7) as well as the structure of GCN2 kinase domain with another ATP-competitive inhibitor[23]. The high structural homology between the PERK and GCN2 kinase domains allowed for unambiguous fitting of

GSK'157 in the ATP-binding pocket of GCN2 (Fig. 4d and Supplementary Fig. 7).

GCN2 has a pseudokinase domain, in addition to its kinase domain, and both domains are present in the active recombinant GCN2 fragment used here. Prior work has firmly established that the mouse pseudokinase domain of GCN2 is inactive and does not bind ATP[24,25]. The pseudokinase domain of GCN2 is highly conserved between mouse and human (Supplementary Fig. 8a), and both lack

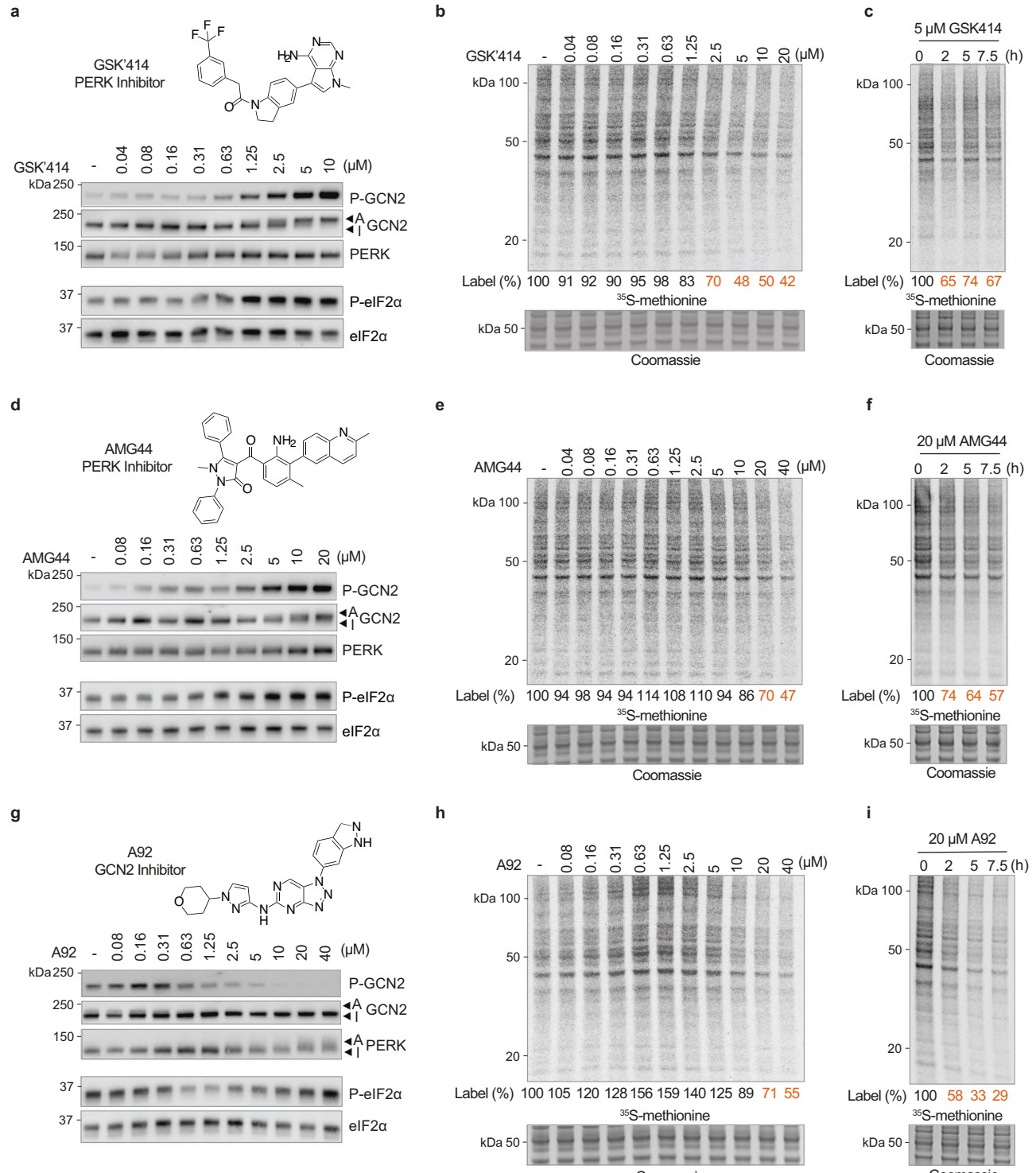

**Fig. 3 | PERK inhibitors activate GCN2 and vice versa. a** Immunoblots of indicated proteins from lysates of HeLa cells treated with indicated concentrations of the PERK inhibitor GSK'414 for 2.5 h. Active (A) and Inactive (I) GCN2 are indicated with arrows. Representative experiment from $n = 3$, biologically independent experiments. **b** Newly synthesized proteins labelled for 10 min with $^{35}S$-methionine in HeLa cells pre-treated with increasing concentrations of GSK'414 for 2.5 h and a Coomassie-stained gel as control. Representative experiment from $n = 2$, biologically independent experiments. **c** Same as panel (**b**), with HeLa cells treated with 5 µM GSK'414 for the indicated times and a Coomassie-stained gel as control. Representative experiment from $n = 2$, biologically independent experiments. **d** Same as panel (**a**), with increasing concentrations of the PERK inhibitor AMG44 for 2.5 h. Active (A) and Inactive (I) GCN2 are indicated with arrows. Representative

experiment from $n = 3$, biologically independent experiments. **e** Same as panel (**b**), with increasing concentrations of AMG44 for 2.5 h. Representative experiment from $n = 2$, biologically independent experiments. **f** Same as panel (**c**), with 20 µM of AMG44 for the indicated times. Representative experiment from $n = 2$, biologically independent experiments. **g** Same as panel (**a**), with increasing concentrations of the GCN2 inhibitor A92 for 5 h. Active (A) and Inactive (I) PERK are indicated with arrows. Representative experiment from $n = 3$, biologically independent experiments. **h** Same as panel (**b**), with increasing concentrations of A92 for 5 h. Representative experiment from $n = 2$, biologically independent experiments. **i** Same as panel (**c**), with 20 µM of A92 for the indicated times. Representative experiment from $n = 2$, biologically independent experiments. Source data are provided as a Source data file.

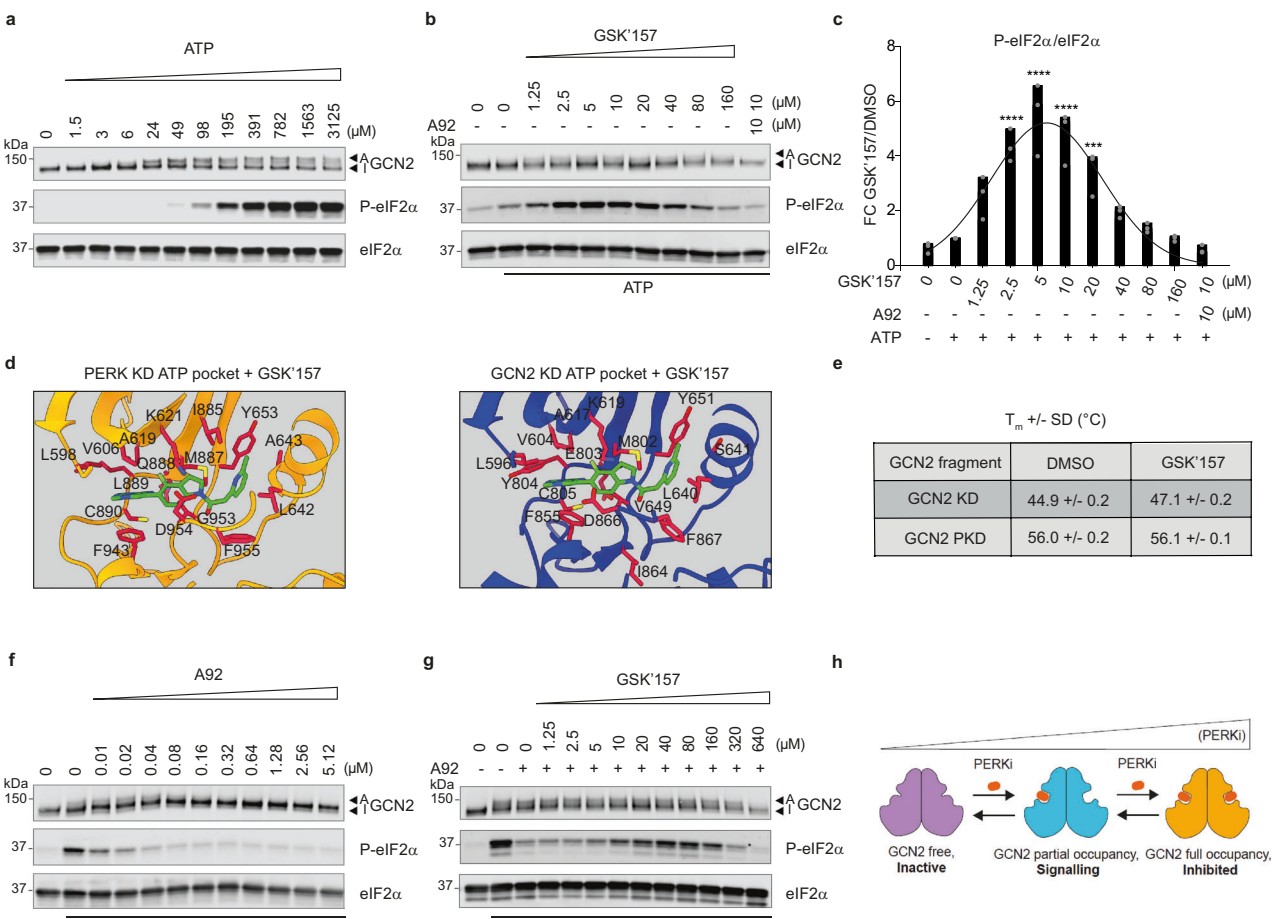

**Fig. 4 | GSK'157 activates GCN2 and induces eIF2α phosphorylation in vitro.**
**a** Immunoblots of indicated proteins from in vitro kinase reactions carried out with 7.5 nM GCN2, 2 µM eIF2α and indicated concentrations of ATP and incubated for 20 min at 30 °C. Active (A) and Inactive (I) GCN2 are indicated with arrows. Representative experiment from *n* = 3, biologically independent experiments.
**b** Immunoblots of indicated proteins from reconstituted kinase reaction as in (**a**), with 6 µM ATP and indicated concentrations of GSK'157, without or with 10 µM of the GCN2 inhibitor A92 compound (last lane). Active (A) and Inactive (I) GCN2 are indicated with arrows. Representative experiment from *n* = 3, biologically independent experiments. **c** Ratio of P-eIF2α to eIF2α levels in immunoblots such as the ones shown in panel (**b**), normalised to 6 µM ATP conditions. Data are shown as mean ± SD (*n* = 3) with a Gaussian curve fitting, biologically independent experiments. ***$p$ = 0.0009, ****$p$ < 0.0001, as determined by one-way ANOVA with Dunnett's multiple comparison test. **d** Structure of PERK kinase domain (KD) with GSK'157 bound to its ATP-binding pocket reported in ref. [11] compared with the structure of GCN2 KD from ref. [23] with GSK'157 fitted into the ATP-binding site.

Yellow–PERK, blue–GCN2, green–GSK'157, red–residues in contact with GSK'157. **e** Melting temperature ($T_m$) of GCN2 kinase domain (KD) or pseudokinase domain (PKD) in the presence or absence of GSK'157 (100 µM) derived from Supplementary Fig. 9. **f** Immunoblots of indicated proteins from in vitro kinase reaction as in (**a**), (**b**), with 12 µM ATP and indicated concentrations of GCN2 ATP-competitive inhibitor A92. Active (A) and Inactive (I) GCN2 are indicated with arrows. Representative experiment from *n* = 2, biologically independent experiments.
**g** Immunoblots of indicated proteins as in (**a**), (**b**), (**f**), from reconstituted in vitro kinase reaction with 12 µM ATP, with or without 0.04 µM A92 and with indicated concentrations of GSK'157. Active (A) and Inactive (I) GCN2 are indicated with arrows. Representative experiment from *n* = 2, biologically independent experiments. **h** Saturating concentrations of ATP-competitive PERK inhibitor (PERKi) occupy both ATP-pockets and inhibit GCN2 whilst sub-saturating concentrations of PERKi result in occupancy of one of the two ATP-binding sites of the GCN2 dimer and kinase activation. Source data are provided as a Source data file.

---

nearly all the invariant kinase residues (Supplementary Fig. 8b). Comparison of the GCN2 kinase and pseudokinase domains also revealed that the pseudokinase domain lacks the residues involved in binding GSK'157 (Supplementary Fig. 8c, d). To experimentally test these structural analyses, we examined binding of GSK'157 to the recombinant GCN2 kinase or pseudokinase domains using thermal shift assays, a standard method to assess compound binding[25]. We found that GSK'157 shifted the melting temperature ($T_m$) of the GCN2 kinase domain but not the pseudokinase domain (Fig. 4e and Supplementary Fig. 9a, b). This reveals that GSK'157 binds the kinase but not the pseudokinase domain of GCN2.

One way to test if GSK'157 activates GCN2 by binding to its ATP-binding pocket is to assess if this compound outcompetes inhibition of GCN2 by the ATP-competitive GCN2 inhibitor A92. Thus, we used our in vitro ISR-activation assay and performed comprehensive dose

response experiments with A92. We found that A92 inhibited GCN2 and eIF2α phosphorylation from 10 nM (Fig. 4f). To evaluate whether GSK'157 can outcompete GCN2 inhibition by A92, we used the minimal concentration of A92 that yielded near complete inhibition of GCN2 in the assay (40 nM) and titrated GSK'157 (Fig. 4g). We found that GSK'157 outcompeted GCN2 inhibition by the ATP-competitive GCN2 inhibitor A92 between 10 and 160 µM (Fig. 4g). At higher concentrations, GSK'157 silenced the kinase (Fig. 4g). The interpretation of the structural insights, binding data, Gaussian activation-inhibition profile, and the competition of ATP-competitive inhibitor A92 with GSK'157 is as follows. GCN2 is active as a dimer and ATP is required for kinase activation. We show that GSK'157 binds the kinase domain and as expected for ATP-competitive inhibitors, the kinase is inhibited at saturating concentrations of the inhibitor (Fig. 4h). This is because the two

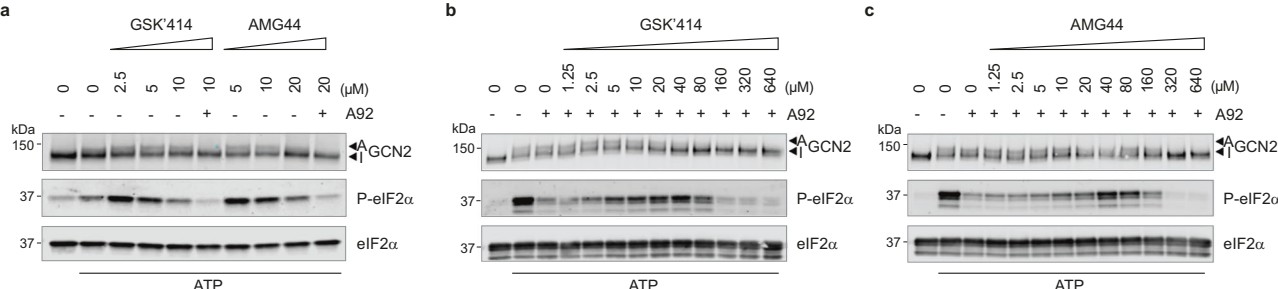

**Fig. 5 | PERK inhibitors of distinct chemotypes activate GCN2 and induce eIF2α phosphorylation in vitro. a** Immunoblots of indicated proteins from in vitro kinase assays with 7.5 nM GCN2, 2 µM eIF2α, 12 µM ATP and increasing concentrations of GSK'414 or AMG44 PERK. 10 µM of the GCN2 inhibitor A92 was added in the indicated lanes. Active (A) and Inactive (I) GCN2 are indicated with arrows. Representative experiment from n = 2, biologically independent experiments. **b**, **c** Immunoblots of indicated proteins as in (**a**), from reconstituted in vitro kinase reactions with 12 µM ATP, with or without 0.04 µM A92 and with indicated concentrations of GSK'414 or AMG44. Active (A) and Inactive (I) GCN2 are indicated with arrows. Representative experiments from n = 2 (per compound), biologically independent experiments. Source data are provided as a Source data file.

ATP-binding pockets of the dimeric kinase are occupied with such ATP-competitive molecules and can no longer bind ATP. Lower (sub-saturating) concentrations of GSK'157 allow reduced occupancy, yielding dimers with one ATP-binding pocket occupied by GSK'157 and the other free to bind ATP, which somehow results in activation of the dimeric kinase (Fig. 4h). The idea that activation of a kinase can be achieved at sub-saturating concentrations by partial occupancy of an inhibitory ligand was first proposed over 10 years ago[26,27]. However, despite the broad relevance of the model, the molecular nature of the activation mechanism of the second protomer remains elusive.

We next tested if the Gaussian activation followed by inhibition of GCN2 by GSK'157 also occurs with other PERK inhibitors. We found that GSK'414 and AMG44, two PERK inhibitors of different chemotypes, also lead to a Gaussian activation profile of GCN2 in vitro (Fig. 5a and Supplementary Fig. 5b). Importantly, GCN2 activation by these PERK inhibitors was inhibited by GCN2 inhibitor A92 (Fig. 5a). Moreover, thermal shift experiments showed that GSK'414 and AMG44 bound the kinase but not the pseudokinase domain of GCN2 (Supplementary Fig. 9a–c). To get further insights into the mechanism of activation, we tested if GSK'414 and AMG44 could alleviate inhibition of GCN2 by ATP-competitive inhibitor A92. Titration of GSK'414 and AMG44 on A92-inhibited GCN2 revealed that the PERK inhibitors outcompeted inhibition of GCN2 by A92 between 2.5 µM and 80 µM for both compounds (Fig. 5b, c). At higher concentrations, GSK'414 and AMG44 inhibited GCN2 (Fig. 5b, c). Thus, this shows that diverse PERK inhibitors activate GCN2 by a similar mechanism.

To further explore the generalizable potential of our findings to diverse ISR kinase inhibitors, we tested if the PKR inhibitor C16[28] could activate GCN2 in vitro. We found that C16 activated GCN2 in vitro between 0.19 and 1.5 µM and inhibited the kinase at high concentrations (Fig. 6a, b). Thus, the PKR inhibitor activated GCN2, following a Gaussian activation-inhibition curve in vitro. C16 also activated GCN2 and the ISR in cells up to 6 µM (Fig. 6c). At 12 µM, ISR induction by C16 appeared reduced, relative to 6 µM (Fig. 6c). Higher concentrations could not be tested in a meaningful way in cells because they resulted in precipitation of the compound in the cell culture media. Importantly, thermal shift experiments revealed that C16, like the PERK inhibitors, bound the kinase domain of GCN2 but not its pseudokinase domain (Supplementary Fig. 9a, b). To gain insights into the mechanism, we tested if C16 could outcompete GCN2 inhibition by the ATP-competitive inhibitor A92, as previously observed for PERK inhibitors. We found that 1.25–40 µM C16 outcompeted GCN2 inhibition by A92, whilst inhibiting GCN2 at concentrations greater than 80 µM (Fig. 6d). This shows that not only diverse PERK inhibitors but also a

PKR inhibitor can activate GCN2 and the ISR through a similar concentration-dependent mechanism.

## GSK'157 activates GCN2 by increasing its affinity for ATP

To understand the mechanistic basis of the described activation, we first examined whether GSK'157 affected GCN2 dimerization, knowing that the kinase is active as a dimer[29]. As previously noted, the recombinant GCN2 used here was dimeric (Supplementary Fig. 4a, b) and GSK'157 had no measurable effect on its dimeric state (Fig. 7a and Supplementary Fig. 10a, b).

To decipher the mechanism by which sub-saturating concentrations of GSK'157 directly activated GCN2, we tested if ADP or AMP-PNP, other binders of the GCN2 ATP pocket[24], could activate the kinase. We found that ADP activated GCN2 and increased phosphorylation of eIF2α between 15 and 250 µM whilst higher concentrations were inhibitory (Supplementary Fig. 11a), resembling the Gaussian activation-inhibition pattern by PERK and PKR inhibitors. Titration of AMP-PNP did not lead to significant kinase activation but inhibited GCN2 from 250 µM (Supplementary Fig. 11b). Thus, distinct kinase ligands show different properties.

Next, we performed kinetic analyses titrating ATP and measuring GCN2 activation (Fig. 7b, c). Addition of GSK'157 increased the velocity of the reaction and dramatically decreased the ATP concentration required to reach half of the maximal velocity (Fig. 7d). These kinetic analyses reveal that GSK'157 increases GCN2 affinity for ATP, thus explaining how it activates the kinase. These results imply that at sub-saturating concentrations, GSK'157 binds to one protomer of the GCN2 dimer, which allosterically transduces a signal to the second protomer and results in its increased affinity for ATP. We next tested the relevance of this mechanism to diverse GCN2 activators. We found that GSK'414 and AMG44 also activated GCN2 by increasing affinity of the kinase for ATP (Fig. 8a, b). The unrelated PKR inhibitor C16 also activated GCN2 by this mechanism (Fig. 8c, d). Recently, a genome-wide knock-out screen revealed that loss of GCN2 rendered cells resistant to the tyrosine kinase inhibitor Neratinib and this occurred by direct activation of GCN2, a property also observed with Dovitinib[30]. We found that Neratinib and Dovitinib both increased the affinity of GCN2 for ATP (Fig. 8e, f), providing a molecular mechanism for their activating properties. Thus, we show that structurally diverse ATP-competitive inhibitors activate GCN2 and the ISR by a common mechanism consisting of increasing the affinity of the kinase for ATP, revealing its broad relevance.

## Discussion

Here we report that the widely used ATP-competitive inhibitor of PERK, GSK'157, inhibited PERK in the nanomolar range as expected[11],

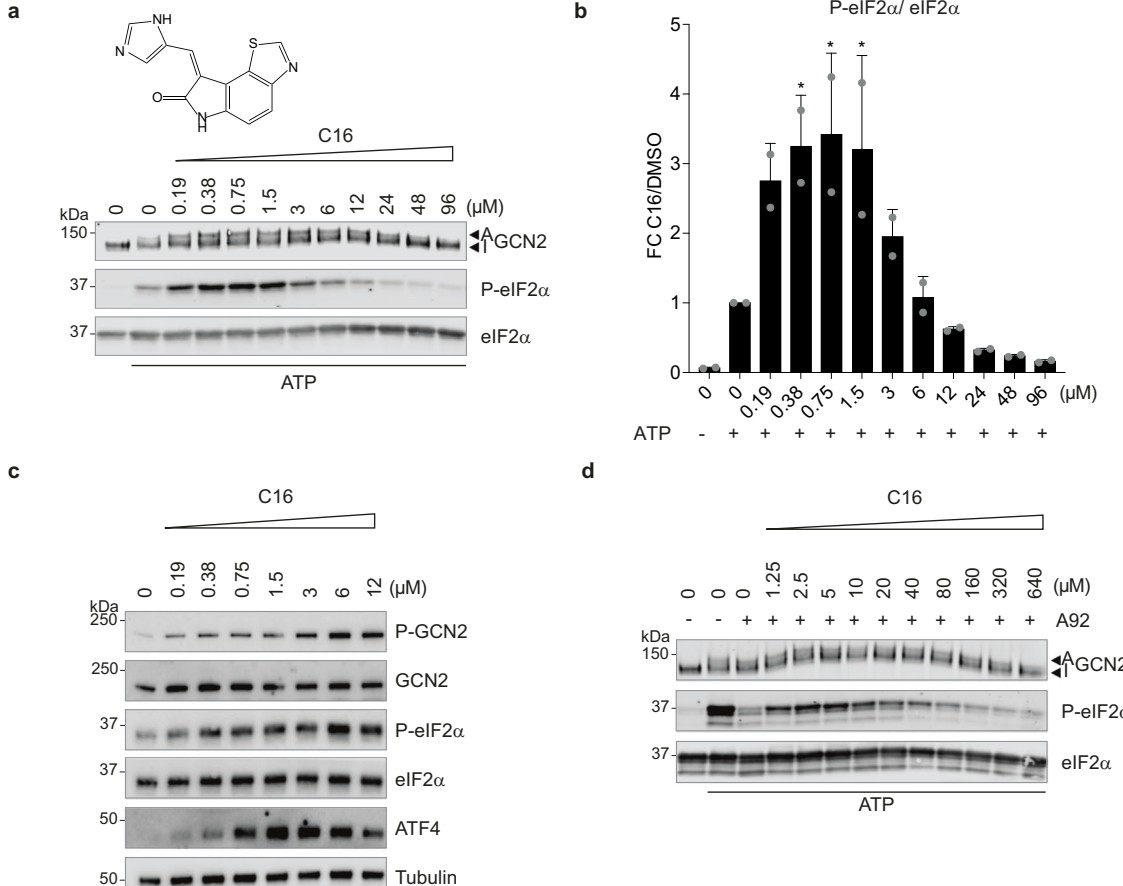

**Fig. 6 | PKR inhibitor C16 activates GCN2 and induces eIF2α phosphorylation in vitro and in cells. a** Immunoblots of indicated proteins from in vitro kinase reaction with 7.5 nM GCN2, 2 μM eIF2α, with or without 6 μM ATP as indicated, and with stated concentrations of C16. Active (A) and Inactive (I) GCN2 are indicated with arrows. Representative experiment from $n = 2$, biologically independent experiments. **b** Ratio of P-eIF2α to eIF2α levels in immunoblots such as the ones shown in (**a**), normalised to 6 μM ATP conditions. Data are shown as mean ± SD ($n = 2$), biologically independent experiments. *$p < 0.03$ as determined by one-way

ANOVA with Dunnett's multiple comparison test. **c** Immunoblots of indicated proteins from lysates of HeLa cells treated with indicated concentrations of the PKR inhibitor C16 for 2.5 h. Representative experiment from $n = 2$, biologically independent experiments. **d** Immunoblots of indicated proteins from reconstituted in vitro kinase reaction as in (**a**), with 12 μM ATP, with or without 0.04 μM A92 and with indicated concentrations of C16. Active (A) and Inactive (I) GCN2 are indicated with arrows. Representative experiment from $n = 2$, biologically independent experiments. Source data are provided as a Source data file.

but surprisingly induced the ISR via GCN2 activation at micromolar concentrations. Similarly, PERK inhibitors GSK'414 and AMG44, as well as the PKR inhibitor C16, activated GCN2 in the micromolar range. Moreover, the GCN2 inhibitor A92 silenced GCN2 activity but induced the ISR via the PERK kinase. Thus, micromolar concentrations of optimized inhibitors of one ISR kinase functionally induce the ISR, rather than inhibit it, by activating a sister ISR kinase (Fig. 9).

The reported findings have broad physiological relevance because PERK inhibitors have been used in over 200 studies as tool compounds to shed light on the function of the ISR, as well as to evaluate the therapeutic potential of its inhibition in various diseases, from cancer to neurodegeneration (https://pubmed.ncbi.nlm.nih.gov/?term=%28%28GSK2606414%29+OR+%28GSK2656157%29%29+OR+%28AMG44%29). Preclinical efficacy studies with PERK inhibitors were conducted at 25 mg/kg or above, a dosing paradigm selected to achieve full inhibition of the target that yielded measured concentrations above 10 μM in plasma, brain, and pancreas[9,12–15]. Here we show that at such concentrations, these compounds inhibit the primary target PERK as expected, yet activate GCN2, resulting in functional ISR activation. The PKR inhibitor C16 has also been used in a number of in vivo studies[31–33]. With the surprising ability of one ISR kinase inhibitor to induce the ISR by activating a related ISR kinase, it will be important to examine what drives the reported efficacy of these

molecules in various disease models. Indeed, many studies intending to inhibit the ISR with ISR kinase inhibitors were conducted at ISR-activating concentrations of the compounds.

Whilst reporting ISR activation with ISR kinase inhibitors is unprecedented, ATP-competitive inhibitors of one given kinase often also inhibit others because of the high-degree of homology between their ATP-binding pockets. Indeed, GSK PERK inhibitors were reported to exert high-affinity inhibitory off-target effects on two other kinases, RIPK1 and KIT[34,35]. However, the selectivity of PERK inhibitors was optimized to reduce potency towards other kinases, and their inhibitory activity towards PERK is at least 3000-fold more potent than for GCN2[9,11,12]. Thus, although some off-target inhibitory activities were reported, it was reasonable to anticipate a relative selectivity of the inhibitor towards the ISR.

Here we discovered that PERK inhibitors exert a distinct type of off-target activity from what was previously anticipated (cross-inhibition) and found that nanomolar affinity PERK inhibitors are micromolar activators of GCN2. This work also provides an explanation to previously confusing observations. The first study reporting on GSK'157 mentioned that "Despite the potent inhibition of PERK activity in the mouse pancreas as shown by decrease in phospho-PERK and histologic changes, we were unable to show inhibition of canonical, downstream PERK signaling."[12]. Two subsequent studies confirmed

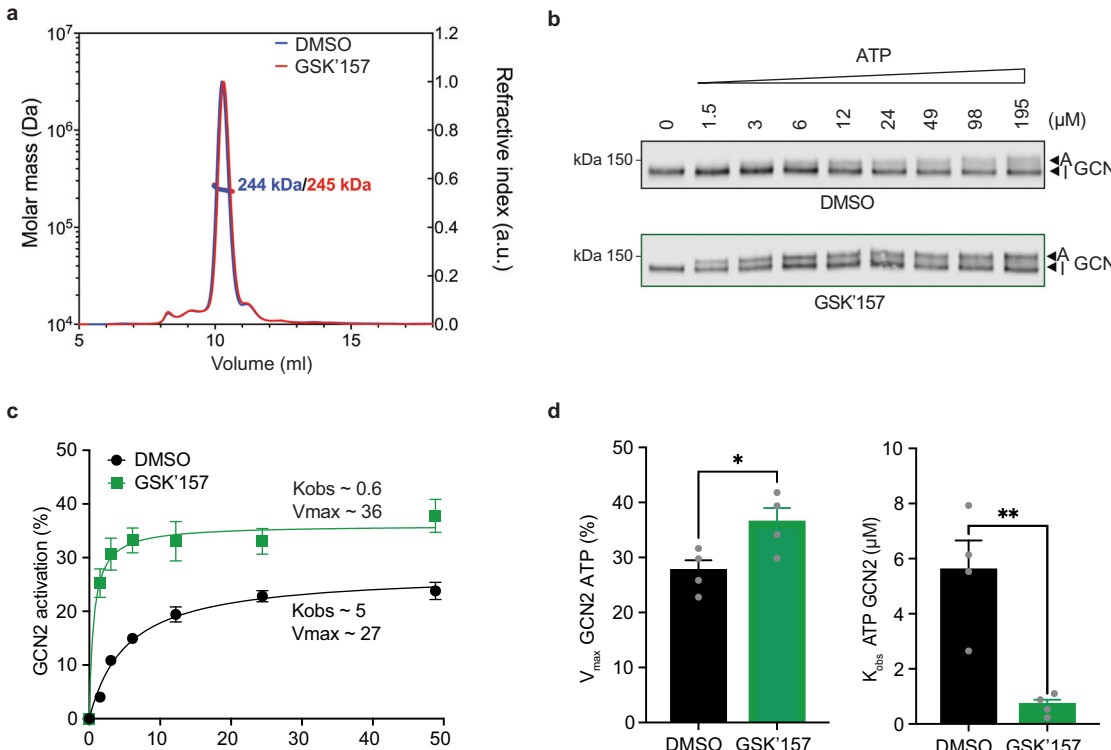

**Fig. 7 | GSK'157 activates GCN2 by increasing its affinity for ATP. a** Size-exclusion chromatography–multiangle light scattering (SEC-MALS) profile of in-house recombinant GST-GCN2 fragment (MW - 135 kDa) in presence of 50 µM GSK'157 or DMSO. The molecular weight measured by light scattering was 244–245 kDa in both conditions, consistent with a dimeric state of the recombinant protein. Representative experiment from $n = 2$, biologically independent experiments. **b** Immunoblots of GCN2 from in vitro kinase reactions with 7.5 nM GCN2 and indicated concentrations of ATP in presence of DMSO or 5 µM GSK'157. Active (A) and Inactive (I) GCN2 are indicated with arrows. Representative experiment

from $n = 4$, biologically independent experiments. **c** Ratio of active to total (active + inactive) GCN2 in the reactions fit to Michaelis-Menten nonlinear model. Data are shown as mean ± SEM ($n = 4$, except 12 µM ATP $n = 3$ for GSK'157), biologically independent experiments. **d** Velocity ($V_{max}$) and apparent affinity ($K_{obs}$) for GCN2 and ATP in presence of DMSO or 5 µM GSK'157 derived from panel (**c**). Data are shown as mean ± SEM ($n = 4$), biologically independent experiments. *$p = 0.0385$, **$p = 0.0046$, as determined by two-sided, unpaired t-test. Source data are provided as a Source data file.

this observation in cells and in vivo[36,37]. Here we found that PERK can be inhibited with GSK'157 in mouse liver, yet the ISR remained activated (Fig. 1a). We recapitulated these in vivo observations in cells and dissected the underlying mechanism using a combination of cellular, genetic and biochemical approaches. Our data reveal that vast excess of an inhibitor saturating the targeted kinase can result in activation of a low-affinity off-target kinase, explaining how an inhibitor of one ISR kinase can bidirectionally control the ISR by activating a sister kinase (Fig. 9). Our findings provide the molecular basis for previously unexplained observations and are pivotal for the use of kinase inhibitors both as tool compounds to probe the function of the ISR and as potential therapeutics in preclinical and clinical studies.

Activation of kinase signalling by a kinase inhibitor was first noted for RAF and IRE1 inhibitors 20 years ago[38,39] and the underlying mechanism involves dimerization of the respective kinases[26,39,40]. More recently, kinase activation by ATP-competitive inhibitors has been reported for PERK and GCN2 occurring through dimerization[27] or stabilization of tRNA binding[30], respectively. These few examples of kinase activation by an ATP-competitive ligand led to the notion that activation can be achieved by partial occupancy of the kinase[26,27]. However, the molecular nature of the activation mechanism of the second protomer has remained elusive. Thus, although there are a few discrete examples of activation of kinases by their inhibitors[41], this notion has not been systematically evaluated.

Activation of GCN2 by GSK'157 occurs in cells in the absence of amino acid stress and this is recapitulated in vitro using a recombinant

GCN2 protein lacking the tRNA binding region of GCN2. Thus, the pharmacological activation of GCN2 reported here by kinase domain-binding compounds bypasses the requirement for natural ligands to activate the kinase. The recombinant GCN2 used here is dimeric and pharmacological activation of GCN2 by ATP-competitive compounds did not change its dimeric status. Activation of the mammalian GCN2 kinase domain has been proposed to involve a switch from an inactive antiparallel dimer to an active parallel dimer[42]. It is possible that the GCN2-activating compounds reported here favour this conformational switch.

Searching for the specific mechanism underlying the pharmacological activation of GCN2 observed here, we discovered that an ATP-competitive inhibitor can activate an off-target kinase by increasing its affinity for ATP. GSK'157 binds the kinase domain of GCN2 and, at saturating concentrations, inhibits the kinase by blocking both protomers, as expected for an ATP-competitive molecule. However, we show that sub-saturating concentrations of GSK'157 directly activate GCN2 by increasing its affinity for ATP. We propose that binding of GSK'157 to one protomer of the dimeric kinase allosterically increases the affinity for ATP of the second protomer. The binding of GSK'157 to one protomer favouring ATP binding to the other reveals positive cooperativity between these events, a mechanism aligned with the classical Monod, Wyman and Changeux allostery model of activation of oligomeric enzymes[43].

Because ATP-competitive kinase inhibitors share the same inhibitory mechanism consisting of occupying the ATP-binding pocket, our findings have broad relevance. ATP-competitive inhibitors used at

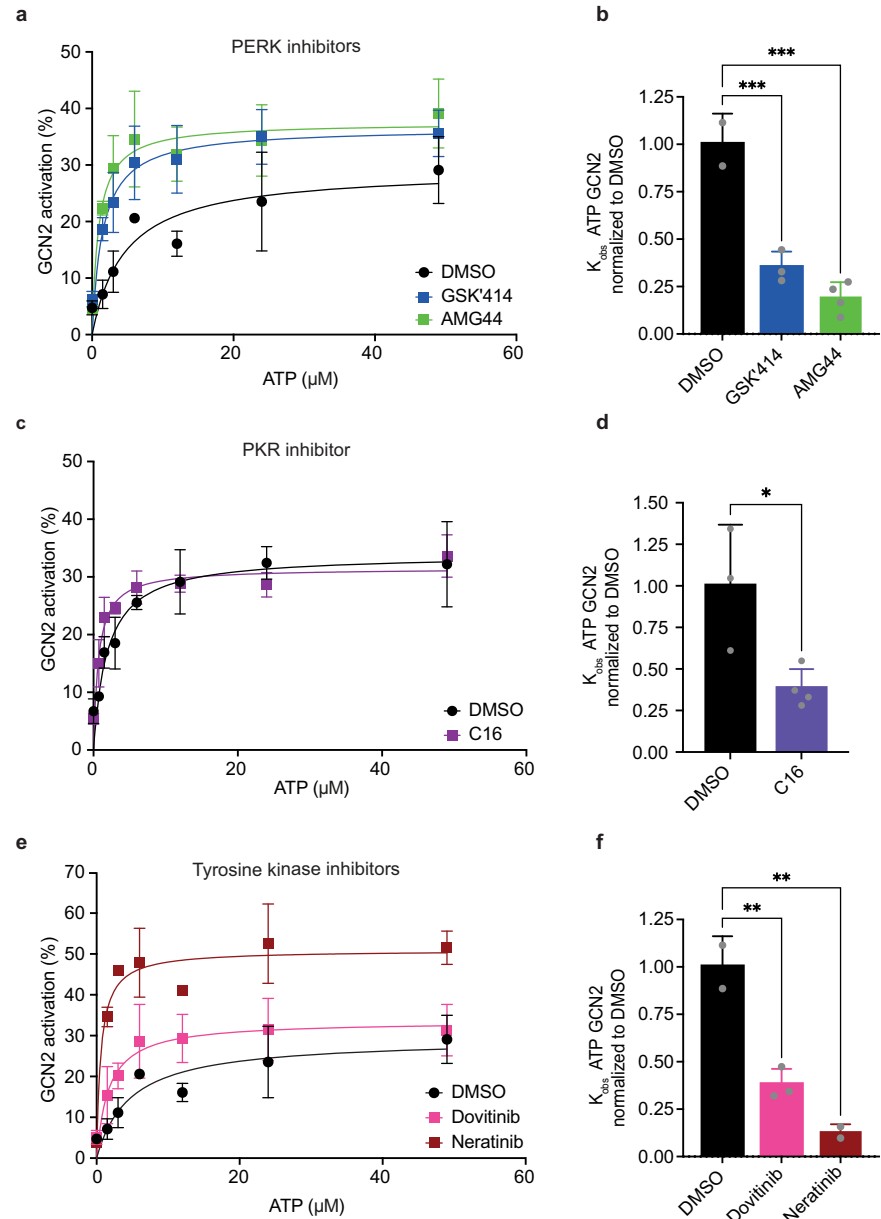

**Fig. 8 | Diverse kinase inhibitors activating GCN2 kinase increase its affinity for ATP. a–f** In vitro kinase reactions with 7.5 nM GCN2 and indicated concentrations of ATP in presence of DMSO or compounds. **a** Ratio of active to total (active + inactive) GCN2 in the reactions with DMSO, 2.5 μM GSK'414 or 5 μM AMG44 fit to Michaelis-Menten nonlinear model. Data are shown as mean ± SD (DMSO $n = 2$, GSK'414 $n = 3$, AMG44 $n = 4$), biologically independent experiments. **b** Apparent affinity ($K_{obs}$) for GCN2 and ATP in presence of compounds from panel (**a**) normalised to DMSO. Data are shown as mean ± SD, biologically independent experiments. \*\*\**p* < 0.0007, as determined by one-way ANOVA with Dunnett's multiple comparison test. **c** As in panel (**a**) for reactions with DMSO or 0.75 μM C16 fit to Michaelis-Menten nonlinear model. Data are shown as mean ± SD (DMSO $n = 3$

and C16 $n = 4$, except 12 μM ATP $n = 2$ and $n = 3$ respectively), biologically independent experiments. **d** As in panel (**b**) for C16 normalised to DMSO. Data are shown as mean ± SD, biologically independent experiments. \**p* < 0.0231, as determined by two-sided, unpaired t-test. **e** As in panel (**a**) for reactions with DMSO, 2 μM Dovitinib or 1 μM Neratinib fit to Michaelis-Menten nonlinear model. Data are shown as mean ± SD (DMSO $n = 2$, Dovitinib $n = 3$, Neratinib $n = 2$), biologically independent experiments. **f** As in panel (**b**) for Dovitinib and Neratinib normalised to DMSO. Data are shown as mean ± SD, biologically independent experiments. \*\**p* < 0.0045, as determined by one-way ANOVA with Dunnett's multiple comparison test. Source data are provided as a Source data file.

concentrations saturating their primary target may activate off-target kinases by the mechanism described here. The markedly different affinities of a compound for the respective ATP-binding pockets of the target versus off-target kinase is a key determinant of its inhibitory versus activating property at a given concentration range. High-affinity binders are biased towards occupancy of both protomers resulting in kinase inhibition while low-affinity binders have a reduced propensity to have both protomers occupied simultaneously, favouring binding to one protomer to potentially activate the other by increasing its

affinity for ATP. Importantly, we find that the mechanism uncovered with the broadly used PERK inhibitor GSK'157 also occurs with other, structurally diverse PERK inhibitors. Further, the same mechanism applies to the PKR inhibitor C16 and the unrelated tyrosine kinase inhibitors Dovitinib and Neratinib, exemplifying the broad relevance of our findings. Importantly, a crystal structure of Dovitinib bound to the kinase domain of GCN2 has been reported showing Dovitinib occupying the ATP-binding-pocket of GCN2, with an active-like conformation of the kinase[42].

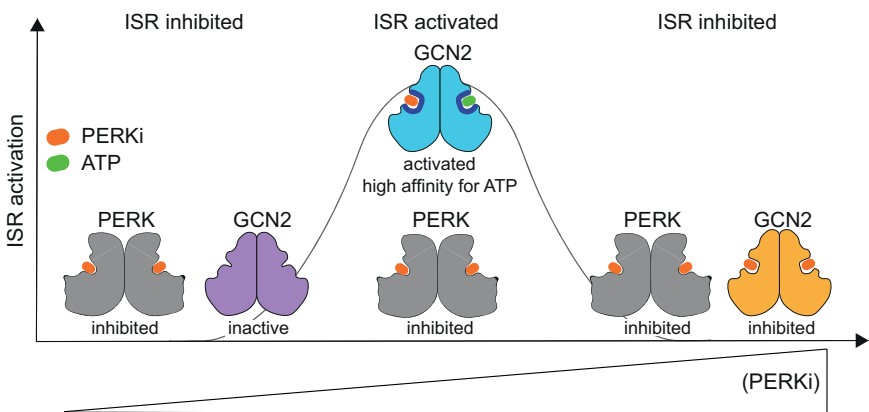

**Fig. 9 | Model for the bidirectional control of the ISR by PERK kinase inhibitors.** At nanomolar concentrations, a PERK inhibitor (PERKi) saturates its primary target and inhibits the ISR, as anticipated[5–7]. At concentrations oversaturating the primary target PERK, binding to the off-target kinase GCN2 is enabled. Binding of the inhibitor to the ATP-binding site of one protomer of the GCN2 dimer, increases the affinity for ATP of the second protomer leading to GCN2 activation. This in turn results in functional ISR induction in absence of specific kinase activating stressors. Saturation of both PERK and GCN2 with complete inhibition of the two kinases is achieved at higher concentrations of the inhibitors. Note that GCN2 and the ISR can be activated by the PKR inhibitor C16 by a similar mechanism.

The mechanism of kinase activation by ATP-competitive inhibitors revealed here made us wonder if PERK inhibitors could activate their primary target when used at low concentrations, before reaching full saturation. This was found to be the case for a GCN2 inhibitor, as reported in a paper published whilst this work was under revision[44]. The concept of activation by substochiometric inhibition has been proposed for proteases, and suggested to be broadly relevant to cooperative enzymes[45]. Here, we observed that low concentrations of GSK'157 in cells caused a mobility shift in PERK, which is usually associated with activation (Fig. 2a). However, this was not accompanied by activation of the ISR (Fig. 2a, b). Likewise, low concentrations of the GCN2 inhibitor A92 increased GCN2 phosphorylation in cell culture without measurable engagement of downstream signalling (Fig. 3g). We concluded that because these inhibitors were selected for high-affinity binding and potent inhibition of their target, they have, by-design, a reduced propensity to robustly activate their primary targets by this mechanism. This explains why the activating properties of ATP-competitive inhibitors reported here are largely biased towards the off-target kinases.

Protein kinase inhibitors represent one of the most prevalent classes of drugs in the pharmaceutical industry[46]. We discovered that a high-affinity ATP-competitive inhibitor of a given kinase acts as an activator of a low-affinity off-target kinase by increasing its affinity for ATP. Because this mechanism is relevant to other kinases, our findings have broad implications. They provide the basis to screen for kinase activators in existing kinase drug screening platforms, to inform on the potential off-target activating properties of ATP-competitive inhibitors, as well as to identify novel molecules. We anticipate this will refine the selectivity of existing protein kinase inhibitors and, therefore, help reduce their clinical side effects. Our findings can also guide the design of the next generation of drugs targeting protein kinases. Thus, the implications of this work, from the mechanism of action of kinase inhibitors to their use in pre-clinical and clinical studies, are broad.

## Methods
### Compounds and reagents
GSK2656157 (#17372), GSK2606414 (#17376), ADP (#16778), Neratinib (#18404) were purchased from Cayman Chemical Company. A92 (#HY-100877) was purchased from MedChemExpress. Dovitinib (#A2168) was purchased from APExBIO. Tunicamycin (#T7765), Imidazolo-oxindole PKR inhibitor C16 (#I9785), AMG44 (#SML3049), EIF2S1 (#SRP5232) and AMP-PNP (#A2647) were purchased from Sigma-Aldrich. ATP (#R0441) was purchased from ThermoFisher

Scientific. EIF2K4 (192-1024) (Active, E12-11G) was purchased from SignalChem Biotech.

### Animal studies
All animal care and procedures were performed in compliance with the regulation on the use of Animals in Research (UK Animals Scientific Procedures Act of 1986 and the EU Directive 2010/63/EU) under the project license number P9DCDB3B0 and with approval from the LMB Animal Welfare and Ethical Review committee. C57BL/6J male mice were used for the assessment of treatment effects. Animal cohort numbers were determined by power analysis based on preliminary results or literature precedent. Mice were housed in pathogen-free ventilated cages (Tecniplast GM500, Tecniplast) on Lignocel FS14 spruce bedding (IPS) and Enviro-Dri nesting material (LBS) at 19–23 °C with 12 h light–dark cycle with light from 7.00 am to 7.00 pm. The number of cages were kept to between 2 to 3 animals per cage. Every day a visual and physical health check of all experimental animals was performed. The experimental animals were weighed prior to the start of the experiment.

### Animal experiments
To evaluate the combined effects of Tunicamycin (Sigma-Aldrich) and GSK2656157 (Cayman Chemical Company) in vivo, 15 mice, C57BL/6J males from 11 to 13 weeks old were used. Mice were injected intra-peritoneally (IP 1) with increasing concentrations of GSK2656157 diluted in DMSO. 30 min after first IP mice were subjected to intra-peritoneal injection (IP 2) with Tunicamycin diluted in $H_2O$. The day before the experiment, animals were weighed and average weight was used to calculate the volume of injections. Mice were culled 2.5 h post second IP by cervical dislocation and exsanguination, and livers were collected and frozen in liquid nitrogen.

### Cell culture
HeLa cells (Sigma-Aldrich, IGBMC, Illkirch, France) were cultured in a humidified incubator with 5% $CO_2$ at 37 °C in Dulbecco's Modified Eagle's Media (DMEM, ThermoFisher Scientific, #D5796) supplemented with 100 U/ml penicillin, 100 µg/ml streptomycin (Thermo-Fisher Scientific, #15140122), 2 mM L-glutamine (ThermoFisher Scientific, #25030081), and 10% fetal bovine serum (ThermoFisher Scientific, #10270106).

For siRNA knockdown experiments, Opti-MEM (ThermoFisher Scientific, #11058021), ON-TARGET plus Human GCN2 (#L-005314-00-0005), HRI (#L-005007-00-0005), PERK (#L-004883-00-0005) and PKR (#L-003527-00-0005) siRNA (Horizon Discovery Biosciences

Limited) and Lipofectamine RNAiMAX (#13778-150, ThermoFisher Scientific) were mixed and incubated for 20 min. Next, the mixes were pipetted to the wells (100 μl/well in 24-well plate) and cells (12,000 cells/well in 24-well plates) resuspended in DMEM were added on top of the transfection mix. After 24 h, media was exchanged to fresh DMEM and after an additional 36 h treatments were initiated.

## Quantitative RT-PCR

RNA was extracted from HeLa cells using RNeasy Mini Kit (QIAGEN) and treated with DNase (RNase-Free DNase Set, QIAGEN) according to the manufacturer's instructions. RNA concentration was measured using a NANODROP1000 spectrophotometer (ThermoFisher Scientific) and 0.5 μg RNA was reverse transcribed to cDNA using iScript cDNA Synthesis Kit (Bio-Rad, #1708891). To evaluate abundance of HRI mRNA, quantitative PCR was performed on a ViiA 7 Real-Time PCR system (Applied Biosystems) using SYBR® Select Master Mix (Applied Biosystems, #4472908) and with the following *HRI* primers: GGAA-GAGTATACAAGGTCAGGAA (F), AGCACCTTCACTTCCCGTAG (R). RNA levels are plotted relative to *β-actin:* GGGCATGGGTCAG AAGGATT (F), TCGATGGGGTACTTCAGGGT (R) and expressed as a fold change.

## Assessment of translation rates

HeLa cells (90,000 cells/ml) were plated in 24-well plates (0.5 ml/well). The next day 0.5 ml of DMEM media supplemented with 2x concentrated compound treatment was added and cells were incubated at 37 °C for the indicated time. After the treatment, cells were washed twice with PBS and labelled with direct addition of 100 μCi/ml $^{35}$S-methionine (Hartmann Analytic) in fresh media (150 μl media with 1.5 μl $^{35}$S-) for 10 min at 37 °C. Note that cells were not cultured in methionine-free media prior to the labelling because methionine depletion is a potent inducer of the ISR pathway. The labelled cells were then washed twice with PBS and lysed in 75 μl Laemmli Buffer (25 mM Tris-HCl pH 7, 7.5% glycerol, 1% SDS, 100 mM DTT, 0.05% bromophenol blue). Lysates were boiled at 95 °C for 5 min, sonicated and resolved on Bolt SDS-PAGE 4–12% Bis-Tris gels (ThermoFisher Scientific). Gels were then stained with Coomassie Blue, destained and imaged to check for equal protein loading. Next, the gels were incubated in 20% ethanol, 7% acetic acid, 4% glycerol solution for 10 min, transferred to filter paper and dried using a gel dryer. Gels were then exposed to a Storage Phosphor Screen (GE Healthcare) and analysed by phosphorimaging using a Typhoon Imager Scanner (GE Healthcare).

## Protein production and purification

**GCN2.** A GST tag was cloned N-terminally into the pAceBac1 GCN2 fragment (192-1024) vector from ref. 48. Before production of baculoviruses and expression in insect cells, the plasmid was integrated into EMBacY baculoviral DNA via Tn7 transposition[48]. Baculoviruses were generated by transfecting the bacmid into Sf9 (ThermoFisher Scientific, Cat#11496015) cells with polyethylenimine (P1) and sequential infections of Sf9 cells to increase the viral titer (P1 → P2, P2 → P3). For protein production, Sf9 cells at $1.8 \times 10^6$ cells/ml were infected with 18 ml of P3 virus per 500 ml of cells. Cells were grown at 27 °C, under shaking conditions (140 rpm) for 55 hours before harvesting, washed in ice-cold PBS and frozen in liquid nitrogen. The cell pellets were thawed and lysed in 50 ml lysis buffer (20 mM Tris-HCl pH 8.0, 150 mM NaCl, 1 mM TCEP, 5% glycerol and 1x protease inhibitor cocktail (Roche Life Science, #48047900) per litre of cells. Benzonase nuclease (Sigma-Aldrich, #E1014) was added to the lysates and the samples were incubated 30 min on ice. Next, lysates were sonicated via a probe sonicator (Sonics Vibra-Cell) for 1 min (10 s on, 10 s off, 70% power). The lysate was then subjected to centrifugation at $100,000 \times g$ for 45 min at 4 °C. The resulting supernatant was added to Glutathione Sepharose 4B (GE Healthcare) equilibrated in lysis buffer. The sample

was incubated for 1 h at 4 °C on rotation. Next, the unbound fraction was collected and the beads were washed in lysis buffer. The protein was eluted with 10 mM reduced L-glutathione (Sigma-Aldrich, #G4251) in lysis buffer at pH 8. The elution was concentrated using a Vivaspin20 30 kDa centrifugal filter and then loaded onto Superdex 200 Increase 10/300 GL column (GE Healthcare) equilibrated in 10 mM HEPES pH 8, 150 mM NaCl and 1 mM TCEP, at a flow rate of 0.5 ml/min. The fractions were resolved on Bolt SDS-PAGE 4–12% Bis-Tris gels (ThermoFisher Scientific), stained with Coomassie Blue and the peak fractions were concentrated using the Vivaspin20, 30 kDa centrifugal filter units before being aliquoted and frozen in liquid nitrogen. Recombinant GCN2 kinase domain and pseudokinase domains were a kind gift from Alison Inglis, Olga Perisic and Roger Williams[47].

**eIF2α.** Full-length human eIF2α (NCBI reference number: NP_004085.1) cloned into the pOPTH vector with an N-terminal His6 tag followed by a TEV protease site was expressed as reported[47].

eIF2α was expressed in One Shot BL21 Star (DE3) Chemically Competent cells (Invitrogen, C601003) in 2x TY media supplemented with 50 μg/ml ampicillin. Cells were grown at 37 °C 220 RPM until OD$_{600}$ 0.5. Expression was then induced for 16 h at 18 °C 220 RPM with 0.1 mM IPTG. Cells were harvested by centrifugation at $3000 \times g$ for 15 min at 4 °C. Pellets were flash-frozen. Frozen pellets were thawed in lysis buffer (20 mM Tris-HCL pH 8, 100 mM NaCl, TBS 1% triton, 1 mM TCEP, 5% Glycerol, 20 mM Imidazole pH 8.0, complete EDTA-free protease inhibitor cocktail (Roche, 11873580001)) using a volume corresponding to 4 times the volume of the pellet. Lysate was sonicated by Sonics Vibra Cell with CV334 converter at 4 °C on ice (10 s on/ 10 s off at 4 °C on ice; 5 min total time; 65% amplification). Lysate was then ultracentrifuged at $21,100 \times g$ for 10 min at 4 °C. Supernatant was next subjected to Ni-NTA agarose beads (QIAGEN, 30210) at 1 ml beads per 7 ml lysate for 2 h rotating at 4 °C. Samples were centrifuged at $600 \times g$ 4 °C for 2 min and flowthrough removed. Beads were next washed 5 times in a BioRad 15 ml gravity column with wash buffer (20 mM Tris pH 8, 100 mM NaCl, 5% Glycerol, 1 mM TCEP, 20 mM Imidazole pH 8.0) before addition of elution buffer (20 mM Tris pH 8, 100 mM NaCl, 5% Glycerol, 1 mM TCEP, 200 mM Imidazole pH 8.0). Peak fractions were combined and diluted 1:1 by buffer Q$_O$ (20 mM Tris pH 8, 5% Glycerol, 1 mM TCEP) to diluted the NaCl to 50 mM. Sample was loaded onto a 5 ml HiTrap Q HP column equilibrated in Q$_A$ buffer (20 mM Tris pH 8, 50 mM NaCl, 5% Glycerol, 1 mM TCEP). The column was washed with 50 ml Q$_A$ buffer and protein was subsequently eluted using a 0–100% gradient of Q$_B$ buffer (20 mM Tris pH 8, 1 M NaCl, 5% Glycerol, 1 mM TCEP). Peak fractions were combined and concentrated to 1 ml before injection onto a HiLoad 16/60 Superdex 75 column. The column was equilibrated with 50 mM HEPES, 150 mM NaCl, 1 mM TCEP and 5% glycerol at a flow rate of 1 ml/min. Peak fractions were concentrated using a centrifugal filter (Sartorius, VS2091) to 2.25 mg/ml (58 μM), aliquoted and flash-frozen in liquid nitrogen for storage at −80 °C.

## In vitro phosphorylation assays

All in vitro phosphorylation reactions were performed in kinase buffer (20 mM Tris-HCl pH 7.4, 150 mM NaCl, 50 mM MgCl$_2$ and 1 mM DTT). 7.5 nM GST-GCN2 fragment (purified in-house or commercial) was incubated for 20 min at 30 °C with ATP (ThermoFisher Scientific), 0.3 μg BSA (ThermoFisher Scientific, #BP1605) per reaction in presence of compounds or DMSO. In assays with eIF2α (initially #SRP5232 from Sigma-Aldrich and then purified in house), the substrate concentration was set to 2 μM. The total volume of the reaction was 15 μl. After incubation the reactions were stopped by addition of 7.5 μl of 4x NuPage LDS Sample buffer supplemented with 200 mM DTT and boiled at 95 °C for 3 min before proceeding with immunoblotting (below).

## Immunoblots and antibodies

**From in cell culture.** HeLa cells (90,000 cells/ml) were plated in 24-well plates (0.5 ml/well) the day before treatment in fresh media to ensure low basal activation of the pathway. The next day 0.5 ml of DMEM media supplemented with 2x concentrated treatment was added and cells incubated at 37 °C for the indicated time. For simultaneous treatments with Tunicamycin and compounds, cells were pre-treated with compounds for 30 min before addition of Tunicamycin. At the end of treatments, cells were washed with ice-cold PBS and lysed in 75 μl Laemmli Buffer (25 mM Tris-HCl pH 7, 7.5% glycerol, 1% SDS, 100 mM DTT, 0.05% bromophenol blue) in situ. Lysates were boiled at 95 °C for 5 min, sonicated and resolved on Bolt SDS-PAGE 4–12% Bis-Tris gels or 3–8% Tris-Acetate gels (ThermoFisher Scientific). Proteins were transferred to nitrocellulose membranes using the Trans-Blot Turbo Transfer System (Bio-Rad). Membranes were blocked in 5% skimmed milk in TBS-T (0.05% Tween) and then probed with primary antibodies diluted in 4% BSA in TBS-T (0.05% Tween): ATF4 (Proteintech, #10835-1-AP, 1:1000), α-tubulin (Sigma-Aldrich, #T5168, 1:5000), p-eIF2α (Abcam, #ab32157, 1:1000), eIF2α (Cell Signaling, #L57A5, 1:1000), eIF2α (Abcam, #ab26197, 1:1000), GCN2 (Cell Signaling, #3302S, 1:1000), p-GCN2 (Abcam, #ab75836, 1:1000), PERK (Cell Signaling, #D11A8, 1:1000), PKR (Santa Cruz, #B-10, 1:1000), Vinculin (Cell Signaling, #4650 S, 1:1000). Next, membranes were washed 3x in TBS-T (0.05% Tween) and incubated with appropriate horseradish peroxidase-conjugated secondary antibodies (Promega, anti-mouse #W402B, 1:5000 and anti-rabbit #W401B, 1:10,000). Proteins were visualized using ECL Prime Immunoblotting System (Cytiva, #RPN2232) and imaged in a ChemiDoc Touch system (BioRad). Note that raw images of typical immunoblots are presented in figures. For detection of total eIF2α and GCN2 proteins immunoblots used for detection of the phosphorylated species were incubated in stripping buffer (ThermoFisher Scientific, #2105) prior application of the total primary antibodies and proceeding with Immunoblotting. Bands were quantified using ImageStudioLite and analyses were performed using GraphPad software.

**From mouse liver.** Liver pieces were lysed in 1 mL of RIPA buffer (1 mM EDTA, 50 mM Tris pH 7.4, 0.25% DOC, 0.1% SDS, 50 mM NaCl, 1% NP40, 2x phosphatase inhibitor tablets (Roche Life Science, #04906837001) and 1x protease inhibitor cocktail (Roche Life Science, #05693159001) using the Percellys bead tissue homogenizer. Extracts were kept 30 min on ice and centrifuged at 18,000 × g at 4 °C for 30 min. Supernatants were collected and protein concentration was measured using the Pierce BCA protein assay (ThermoFisher Scientific, #23225). Extracts were adjusted to the same protein concentration and 50 μl of 4x NuPage LDS sample buffer (ThermoFisher Scientific) with 200 mM DTT was added to every 100 μl of samples and boiled at 95 °C for 5 min. Equal volumes of extracts were loaded on Bolt SDS-PAGE 4–12% Bis-Tris gels or 3–8% Tris-Acetate gels (ThermoFisher Scientific). Proteins were transferred to nitrocellulose membranes using the Trans-Blot Turbo Transfer System (Bio-Rad). Membranes were blocked in 5% skimmed milk in TBS-T (0.05% Tween) and then probed with following primary antibodies diluted in 4% BSA in TBS-T (0.05% Tween): α-tubulin (Sigma-Aldrich, #T5168, 1:5000), p-eIF2α (Abcam, #ab32157, 1:1000), eIF2α (Abcam, #ab26197, 1:1000) and PERK (Cell Signaling, #C33E10, 1:1000). Next, membranes were washed 3x in TBS-T (0.05% Tween) and incubated with appropriate horseradish peroxidase-conjugated secondary antibodies (Promega, anti-mouse #W402B, 1:5000 and anti-rabbit #W401B, 1:10,000) and immunoblots were performed as described above. Bands were quantified using ImageStudioLite and analyses were performed using GraphPad software.

**From in vitro experiments.** Equal volumes (5 μl) of the in vitro reactions were loaded on 3–8% Tris-Acetate gels (ThermoFisher Scientific). Proteins were transferred to nitrocellulose membranes using the Trans-Blot Turbo Transfer System (Bio-Rad). Membranes were blocked in 5% skimmed milk in TBS-T (0.05% Tween) and then probed with following primary antibodies diluted in 4% BSA in TBS-T (0.05% Tween): p-eIF2α (Abcam, #ab32157, 1:1000), eIF2α (Cell Signaling, #L57A5, 1:1000), GCN2 (Cell Signaling, #E9H6C, 1:1000). Next, membranes were washed 3x in TBS-T (0.05% Tween) and incubated with appropriate secondary antibodies: Goat anti-Mouse Alexa Fluor 680 (Invitrogen, #A32729, 1:5000) or Goat anti-Rabbit Alexa Fluor 790 (Invitrogen, #A27041, 1:10,000). Proteins were visualized using the LICOR Odyssey infrared imaging system. The detection of p-eIF2α and eIF2α proteins was performed simultaneously. Bands were quantified using ImageStudioLite and analyses were performed using GraphPad software.

## Size exclusion chromatography-multiangle light scattering (SEC-MALS)

Size-exclusion chromatography coupled with multiangle static light scattering (SEC-MALS), was performed using an Agilent 1200 series LC system with an online Dawn Helios ii system (Wyatt) equipped with a QELS+ module (Wyatt) and an Optilab rEX differential refractive index detector (Wyatt). GST-GCN2 fragment purified in-house was incubated for 20 min at 30 °C in presence of DMSO or 50 μM GSK'157. 100 μl purified protein at ~1.7 mg/ml was auto-injected onto a Superdex 200 Increase 10/300 GL column (GE Healthcare) and run at 0.5 ml/min. The system was equilibrated with 10 mM HEPES, 150 mM NaCl, 1 mM TCEP with DMSO or 50 μM GSK'157 depending on the analysis. The instrument was calibrated using Bovine Serum Albumin (ThermoFisher Scientific, #23209) as a standard. The molecular masses were analysed with ASTRA 7.3.0.11 (Wyatt) and data were plotted using GraphPad software.

## Intrinsic thermal shift

The Prometheus NT.48 instrument (NanoTemper Technologies) was used to determine melting temperatures of GCN2 domains with and without compound addition. 5 μM of His-GCN2 KD (585-1024) or His-GCN2 PKD (192-539) were used per condition in 20 mM HEPES pH 7.5, 30 mM $MgCl_2$ and 150 mM NaCl. Samples were incubated with indicated compound, using concentrations adjusted to add 1 μl of compound or DMSO to make a total volume of 25 μl. The samples were incubated for 10 min at RT before capillaries were filled and placed onto the Prometheus sample holder. A temperature gradient of 2 °C per min from 15 to 95 °C was applied and the intrinsic protein fluorescence at 330 and 350 nm was recorded. The first derivative ratio (330 nm/350 nm) was calculated by the Prometheus NT.48 instrument software. Graphs were plotted using Prism 9.4.1 (GraphPad Software, Inc).

## Quantification and statistical analysis

Sample size for each experiment was determined based on previous studies.

The statistical analysis was performed using Prism 9.4.1 (GraphPad Software, Inc) using unpaired t-test, one-way ANOVA or nonlinear regression curve with variable slope as indicated in the figure legends. The data are presented as mean ± SD for $n < 4$ and ±SEM for $n \geq 4$.

## Reporting summary

Further information on research design is available in the Nature Portfolio Reporting Summary linked to this article.

## Data availability

Data generated or analysed during this study are either included in this article, its Supplementary materials, and the source data file or available from the corresponding author upon request. Source data are provided with this paper.

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

## Acknowledgements
We thank Shashank Rai and LMB Animal Facility for help with animal experiments; Ketan Malhotra for help with protein purification; Stephen McLaughlin for help with SEC-MALS and thermal shift experiments; Alison Inglis, Olga Perisic and Roger Williams for GCN2 and eIF2α plasmids, as well as, recombinant GCN2 pseudokinase and kinase domains; Michel Goedert and Petra Gross for insightful comments on the manuscript; Emily Naden for proof-reading the paper. This work was supported by Medical Research Council (UK) grant MC_U105185860 and Wellcome Trust Principal Investigator Award 206367/Z/ 17/Z to A.B. M.S. was supported by Human Frontier Science Program (HFSP) LT000162/ 2021-L and European Molecular Biology Organization (EMBO) ALTF 698-2020.

## Author contributions
A.B. and M.S. designed the study and experiments. M.S., D.A.J., C.M. and G.H. conducted and analysed experiments with input from A.B. M.S. and A.P.P. conducted protein translation experiments. M.S. and A.F. expressed and purified the GCN2 protein. G.H. and A.F. expressed and purified eIF2α. M.S. and D.A.J. designed, conducted and analysed the in vitro experiments. C.M. designed and conducted the experiments with C16 and the competition experiments with A92. G.H. designed and conducted the thermal shift experiments. A.A. did the sequence alignments and molecular dockings. M.S., C.M., G.H., A.A. and A.B. prepared the figures. A.B. wrote the manuscript with M.S. and input from all the authors. A.B. supervised the study, guided its design and implementation and provided funding.

## Competing interests
The authors declare no competing interests.
