## [Peer Review File · Nature Communications]

REVIEWER COMMENTS

Reviewer #1 (Remarks to the Author):

Szaruga and colleagues investigated the mechanism of induced phosphorylation of the translation initiation factor eIF2 (p-eIF2) upon treatment with inhibitors of the eIF2 kinases PERK and GCN2. It has been known for a long time (references within the manuscript) that PERK or GCN2 inhibitors can stimulate p-eIF2 in cells and mice after prolonged treatments as well as treatments in high concentrations. This is confirmed by the authors in Fig. 1 of the manuscript. However, the mechanisms of increased p-eIF2 by the eIF2 kinase inhibitors are not well understood. The authors propose a model in which low concentrations of PERK inhibitors stimulate GCN2 activity upon binding to the kinase with low affinity and increasing its affinity for ATP whereas higher concentration of the PERK inhibitors lead to inhibition of GNC2. This model is mainly proposed by the in vitro assays with purified proteins. To begin with, the data with mice and cells indeed demonstrate the ability of PERK or GCN2 inhibitors to stimulate the activity of GCN2 and PERK, respectively. But it is unclear whether such an effect is fully explained by the proposed model or regulation of pathways such as the expression and function of phosphatases that target the eIF2 kinases. The in vitro data is as good as it can get. The limitation is the source of purified GCN2 since the GST component of it can drastically affect its activity in vitro and interpretation of the data. One important question is how purified GCN2 functions in the presence of the inhibitors when its natural ligand (uncharged tRNA) is added to reactions. It is not clear either what the source of purified eIF2 is. Control experiments testing eIF2 phosphorylation in the absence of GCN2 should be included to exclude the possibility of phosphorylation by contamination of eIF2 with eIF2 kinases. It is of interest that ADP is more potent in activating GST-GCN2 than AMP-PNP even in the presence of 10x higher concentration of each compound than ATP (Suppl. Fig. S8). One would expect that the high concentration of each compound should result in full occupancy of GCN2 and inhibition of its activity (at least for AMP-PNP). Thus, this Reviewer feels that the model requires thorough re-evaluation..

Reviewer #3 (Remarks to the Author):

The Integrated Stress Response (ISR) is a central homeostatic network that maintains cellular health. Over the last few years, dysregulation of the ISR has been associated with neurodegeneration and cognitive disorders. Therefore, the ISR emerges as an attractive target for pharmacological intervention in human disease. In this study, Szaruga et al., reported that commercially available small molecules designed to specifically inhibit the ISR sensor PERK activate the ISR kinase GCN2 at high concentrations. By combining in vivo studies with biochemical assays, the authors showed that the GSK's PERK inhibitors stimulate GCN2 activity both in vitro and in vivo. Similarly, the GCN2 inhibitor A92 activates PERK at a high dosage.

Overall, the manuscript is well-written, the experiments are high quality. However, some of the authors' claims are not fully supported by the data and additional experiments are required to support their conclusions. Here below there are a few points that should be addressed.

My major concern is that the evidence provided to support the model of activation of GCN2 by GSK'157 is weak. Several other possibilities could explain the authors' conclusions.

1) The authors should test kinase extrinsic and intrinsic mechanisms. Is there an inhibitor-induced feedback loop? To test this possibility, the authors should test the effect of the inhibitors in cells in which eIF2 cannot be phosphorylated. As per kinase intrinsic mechanism, the authors should test for changes in the kinase itself, including the use of catalytically dead mutant PERK or mutants in which the inhibitor cannot bind (Shokat method).

2) Does high concentrations of GSK'157 increase ER membrane localization of PERK?

3) Kinetic analysis shows that GSK'157 lowers the threshold of activation of GCN2. The authors interpreted this finding as reasoning that GSK'157 is an allosteric GCN2 activator. They propose that GSK binds to GCN2 protomer in the ATP pocket favoring ATP binding to the other protomer. However, GCN2 forms constitutive dimers in an inactive antiparallel conformation. Activation of the kinase requires a conformational change that re-organizes GCN2 dimers into a parallel active state. Thus, a possible scenario is that GSK compounds stabilize GCN2 in a parallel conformation.

4) Additionally, the ISR sensor kinases tend to form oligomers to regulate their activity. Another non-mutual exclusive possibility is that GSK'157 favors GCN2 high-order assembly increasing the activity.

5) Finally, GCN2 contains a pseudokinase domain (YKD) that is required to stabilize and boost kinase activity. The recombinant protein used in this study for the in vitro kinase assay lacks the t-RNA binding domain but maintains the YKD. In this scenario, GSK'157 might bind to the YKD stabilizing KD-YKD interactions.

The authors should do additional experiment to rule in or out the different scenarios described above.

6) Treatment with the GCN2 inhibitor A92 seems to induce PERK at a high dosage. Does A92 activate the UPR? PERK and PKR have a high degree of homology. To claim generality of the process, the authors

should test at least an additional ISR inhibitor (C16) and see whether the same behavior takes place. Moreover, does A92 show activity towards PKR? Measuring the effect of A92 on the other ISR kinases would strengthen the findings.

Other points

1) The in vivo experiment convincingly shows that the administration of tunicamycin in the presence of the PERK inhibitor GSK'157, blocks PERK activity (i.e., autophosphorylation) but does not have any effect on the phosphorylation of the PERK substrate eIF2 α . To gain insight into these conflicting results, the authors used cell culture to test the efficacy of GSK'157 in blunting ISR signaling upon tunicamycin treatment. Dose-response analysis revealed that GSK'157 activates the ISR at a high dose. To justify their focus on GCN2, in lines 23-30 of page 3, the authors state "Searching for the molecular basis of this phenomenon, we reasoned that PERK inhibition might overwhelm another eIF2 α kinase. Accumulation of misfolded proteins during unresolved ER stress could prevent their degradation and thereby diminish the pool of amino acids, a signal to activate GCN2, the amino-acid sensing ISR kinase^{17,18}. We found that, in Tm-stressed cells, ISR-activating concentrations of GSK'157 caused robust activation of GCN2 detected by its phosphorylation and reduced mobility on SDS-PAGE (Fig. 1f). This demonstrates that, in the presence of ER stress, the benchmark PERK inhibitor GSK'157 activates GCN2 and the ISR in the low micromolar range."

2) Tunicamycin is a pleomorphic drug that activates PKR (PMID: 19007793). To my knowledge, there is no formal evidence that the inhibition of PERK upon UPR induction is compensated by GCN2 and the experiments showed in figure 1 do not rule out this possibility. Therefore, the conclusion of the first paragraph should be rephrased.

Moreover, the manuscript would benefit by showing the activity (i.e. autophosphorylation) of all ISR kinases in cells co-treated with tunicamycin and increasing concentration of GSK'157.

3) Figure 2C. The blot shows high levels of p-eIF2 α at a steady state (line 1). Has the control been treated with a cationic lipid transfectant reagent? One possibility might be that the transfection/handling activated the ISR. While the p-eIF2 α WB might lead to misinterpretation, ATF4 expression results are very clear. The authors should consider replacing the panel.

4) Figure 2D. p-eIF2 α levels in the blot relative to PERK KD are hard to see due to a technical artefact. Please replace the image.

5) Figure S2. Validation of KD efficiency for GCN2, PKR, and PERK is missing.

6) Figure 3G. The experiment shown in figure 3G indicates that the GCN2 inhibitor A92 activated PERK at a high dosage. The activity of A92 on GCN2 was assayed in cells in steady-state conditions since p-GCN2 was already visible in the absence of stress. The authors reasoned that the p-GCN2 observed is due to kinase basal activity. If this is true, then why p-GCN2 basal levels are undetectable in Figures 3A and

The paper by Szaruga et al attempts to understand an interesting observation that the authors make that certain PERK inhibitors activated the integrated stress response (ISR). ISR signalling is dependent upon 4 kinases that detect different signalling inputs -- this leads to eIF2a phosphorylation and subsequent brief protein translational attenuation. The two PERK inhibitors GSK157 and GSK 414 were seen to inhibit PERK but activate GCN2 in cells. In vitro experiments were used to further dissect the mechanism and showed that in the presence of ATP, GSK157 activated GCN2 to phosphorylate eif2A and this followed a gaussian distribution. The authors put forth a mechanistic model based on this.

On the whole, the experiments are done well and that the data shows an interesting observation in cells. This work helps (at least partially) to understand observations where addition of inhibitors does not lead to downstream ISR signalling inhibition. However, there are major points the authors need to address - The conclusions/claims of the paper particularly the model of activation is not supported by the data in my opinion and authors should either move this to the discussion along with adjusting their claims (this will affect the impact of the paper) or provide the necessary data.

1)The purified protein construct encompasses both the kinase domain and the pseudo-kinase domain. The interpretation of results is completely devoid of the mechanistic implications of pseudo-kinase domain. It important to note that PERK does not have a pseudo-kinase domain (or any of the auto-inhibitory domains that GCN2 has). The authors do not demonstrate that the inhibitors bind to GCN2 ATP active site but assume this to be true based on its similarity to PERK. It could be that the inhibitors bind to pseudo-kinase site or another site on the protein and that this allosterically activates GCN2 kinase activity. Whilst the pseudo kinase domain may not bind ATP due to the lack of Mg²⁺ coordination, it could bind the inhibitors. Allosteric regulation of kinases by the pseudo-kinase domain is well known although not completely understood. A previous study has shown GCN2 pseudo kinase domain regulation of kinase activity (PMID 24811037) leads to enhanced eif2a activation. Thus, the authors should demonstrate if the inhibitors bind to the active site or the pseudo-kinase domain or both and include pseudo-kinase regulation of kinase activity in their assessment and interpretation.

2)what is the affinity of GSK157 and GSK414 to GCN2? This is important information to understand what a sub-saturating concentration is?

3)In the invitro analysis the concentration of purified GCN2 is at 7.5nM, while the addition of ATP is at 6µM. This is almost 1000 times excess of ATP. Similarly, the inhibitor concentration is micro molar range. The authors do not provide the affinity of the inhibitors to GCN2 but assuming that it maybe be close to that of PERK or even 10-fold less, this would still suggest all binding sites are saturated even at their lowest concentration of inhibitors. Thus, the authors need to clarify why they suggest that this is sub saturating conditions?

4)Following on from the above point – what is the affinity of ATP to GCN2? If ATP is in excess this would out compete the non-specific PERK inhibitor? This makes it more important to state what the binding affinity is for the inhibitors to GCN2.

5) In Fig 4a – the ATP concentration is listed in mM but in fig 4b and fig 4c the inhibitor concentrations are listed as μM . They should change the fig 4a so that the ATP concentration is in μM same as fig 4b and fig 4c.

6) In Fig 2d – there is no measurement of HRI protein levels.

7) The amount of protein expressed in cells of PERK and GCN2 may differ thus small amounts of inhibitor may actually be saturating for PERK while sub-saturating for GCN2. The implications of this should be mentioned in the discussion.

Other minor points:

8) The authors make a link between low concentration of inhibitor and that this corresponds to only one molecule in a dimer having inhibitor bound. What evidence do they have to state this? Why could not low levels of inhibitor bind to both protomers in a dimer of GCN2 and then not bind many other GCN2 dimers?

9) I am not entirely sure what the point of the gaussian curve is as it will change if you increase any of the limiting factors i.e. if you add more ATP then the curve will shift?

10) In certain figures the difference between activated and inactivated bands are very difficult to interpret. Why not run the gel for longer and thus allow greater separation between active and inactive bands?

Reviewer's Comments:

Reviewer #1 (Remarks to the Author)

Szaruga and colleagues investigated the mechanism of induced phosphorylation of the translation initiation factor eIF2 (p-eIF2) upon treatment with inhibitors of the eIF2 kinases PERK and GCN2. It has been known for a long time (references within the manuscript) that PERK or GCN2 inhibitors can stimulate p-eIF2 in cells and mice after prolonged treatments as well as treatments in high concentrations. This is confirmed by the authors in Fig. 1 of the manuscript. However, the mechanisms of increased p-eIF2 by the eIF2 kinase inhibitors are not well understood. The authors propose a model in which low concentrations of PERK inhibitors stimulate GCN2 activity upon binding to the kinase with low affinity and increasing its affinity for ATP whereas higher concentration of the PERK inhibitors lead to inhibition of GNC2. This model is mainly proposed by the in vitro assays with purified proteins.

To begin with, the data with mice and cells indeed demonstrate the ability of PERK or GCN2 inhibitors to stimulate the activity of GCN2 and PERK, respectively. But it is unclear whether such an effect is fully explained by the proposed model or regulation of pathways such as the expression and function of phosphatases that target the eIF2 kinases. The in vitro data is as good as it can get.

Our response:

We thank the reviewer for appreciating the value of the in vitro data.

One statement from the reviewer is incorrect "This model is mainly proposed by the in vitro assays with purified proteins." The activation of GCN2 by PERK inhibitors was not only recapitulated in vitro but also observed in cells in a series of experiments and using a comprehensive combination of readouts (ISR markers by immunoblots and translation assays) in various conditions (cell exposed to stress or not) and following knockdown of ISR kinases. It is only after the experiments in cells ruled out the possibility that activation of the ISR occurs by any other mechanism (phosphatases or else) than GCN2 activation that we set out to reconstitute this in vitro to dissect the underlying mechanism because detailed molecular mechanisms cannot be unambiguously elucidated in the complex cellular environment.

The key findings are summarized here:

1. Figure 1 shows that the PERK inhibitor GSK'157 inhibits PERK and the ISR in the nanomolar range as anticipated but surprisingly activates the ISR in the low micromolar range in ER stressed cells. At this point, we thought just as the reviewer that any mechanism, direct or indirect could account for the results.
2. We in fact initially favored the idea of an indirect mechanism, just like the reviewer. As discussed in the manuscript, we "*reasoned that PERK inhibition might overwhelm another eIF2 α kinase. Accumulation of misfolded proteins during unresolved ER stress could prevent their degradation*

and thereby diminish the pool of amino acids, a signal to activate GCN2, the amino-acid sensing ISR kinase^{17,18}.” This led us to find that, “in *Tm*-stressed cells, ISR-activating concentrations of GSK’157 caused robust activation of GCN2 detected by its phosphorylation and reduced mobility on SDS-PAGE (Fig. 1f). This demonstrates that, in the presence of ER stress, the benchmark PERK inhibitor GSK’157 activates GCN2 and the ISR in the low micromolar range.”

3. At that point of the work, it was not clear how GCN2 was activated, and we again favoured the idea of an indirect mechanism. We then went on testing “whether the ISR-activating property of GSK’157 was dependent on ER stress.” This was not the case, as shown in Figure 2. We observed that “micromolar concentrations of the PERK inhibitor GSK’157 specifically activate GCN2 and induce a functional ISR in cells independently of ER stress or PERK.”

This narrowed down the search and suggested that “GSK’157 activates GCN2 independently of PERK. Indeed, GSK’157 activated GCN2 and induced functional ISR in cells after PERK had been knocked down (Fig. 2c).”

4. Importantly, we continued our search and “we knocked-down each ISR kinase in HeLa cells (Fig. 2d and Supplementary Fig. S2). Inactivation of GCN2 but not HRI, PERK or PKR abolished the increase of eIF2 α phosphorylation and ATF4 induction by GSK’157 (Fig. 2d).” **This demonstrates that ISR activation by GSK’157 is entirely mediated by GCN2.** We emphasised this conclusion in the revised manuscript: this finding is very important because it rules out the possibility of any alternative mechanism.

1) Reviewer #1 The authors propose a model in which low concentrations of PERK inhibitors stimulate GCN2 activity upon binding to the kinase with low affinity and increasing its affinity for ATP whereas higher concentration of the PERK inhibitors lead to inhibition of GNC2.

Our response:

This statement, together with the previous one “This model is mainly proposed by the in vitro assays with purified proteins.” raised a good point. We did not show the full dose response of PERK inhibitor in cells. This has now been added. Having found the Gaussian activation-inhibition curve in vitro, we then went back to cells and tested a larger dose response. We found that the Gaussian activation-inhibition curve of GCN2 by GSK’157 was recapitulated in cells (Supplementary Fig. S6). This validated in cells the mechanism revealed in vitro with recombinant proteins. We feel that this is an important addition and thank the reviewer for this suggestion.

2) Reviewer #1 The limitation is the source of purified GCN2 since the GST component of it can drastically affect its activity in vitro and interpretation of the data.

One important question is how purified GCN2 functions in the presence of the inhibitors when its natural ligand (uncharged tRNA) is added to reactions.

Our response:

The source of the recombinant GCN2 is not a limitation but a strength. Yet, the reviewer makes an important point here, as our findings were insufficiently explained.

The activation of GCN2 by its natural ligands (uncharged tRNA) is an interesting and important question but it is not relevant to the present study. The detailed reasons for this are explained below.

In cells, GCN2 is activated by the PERK inhibitors in absence of amino acid stress, demonstrating that this chemical activation of GCN2 bypasses the requirement of its natural ligands. This is an important finding that we now emphasize in the revised manuscript. Because the chemical activation of GCN2 occurs in cells in absence of stress, the activation of GCN2 by the PERK inhibitors was tested *in vitro* using a kinase fragment characterized by others to be functional yet lacking regulatory domains of the kinase such as the tRNA binding region. We found that the PERK inhibitors activate the GCN2 fragment lacking the tRNA-binding region: this demonstrates that the PERK inhibitors activate GCN2 independently of tRNA binding. This explains why in cells, activation is observed in absence of stress. Thus, there is no need to incorporate the tRNA sensing domain nor tRNA in our reactions because chemical activation of GCN2 bypasses the requirement of the natural ligands. Of note, as observed here pharmacologically, mutants in yeast GCN2 were found to activate the kinase without the requirement of natural ligands (Padyana, A. K., Qiu, H., Roll-Mecak, A., Hinnebusch, A. G. & Burley, S. K. Structural Basis for Autoinhibition and Mutational Activation of Eukaryotic Initiation Factor 2 α Protein Kinase GCN2*[boxes]. *J Biol Chem* **280**, 29289–29299 (2005).) This was not sufficiently emphasized in the manuscript and has been rectified in the revised version.

We thank reviewer for his/ her comment on the *in vitro* assay. *In vitro* reconstitution of cellular events is difficult and sometimes takes decades. For example, the Raf inhibitors were reported to activate Raf signaling in 1999 (Hall-Jackson, C. A. *et al.* Paradoxical activation of Raf by a novel Raf inhibitor. *Chem Biol* **6**, 559–568, (1999) and the molecular basis for this proposed using cell-based readouts more than 10 years later (Hatzivassiliou, G. *et al.* RAF inhibitors prime wild-type RAF to activate the MAPK pathway and enhance growth. *Nature* **464**, 431–435 (2010); 1.Poulikakos, P. I., Zhang, C., Bollag, G., Shokat, K. M. & Rosen, N. RAF inhibitors transactivate RAF dimers and ERK signalling in cells with wild-type BRAF. *Nature* **464**, 427–430 (2010).). So far, Raf activation has not been reconstituted *in vitro* with recombinant proteins.

Recapitulating the activation of GCN2 by the PERK inhibitors *in vitro* is an important achievement. As explained above, we embarked on this work only after our thorough and comprehensive experiments in cells revealed that activation of the ISR by the PERK inhibitors was entirely mediated by GCN2. The knockdown of GCN2 abolished ISR activation by the PERK inhibitors, revealing that there are no alternative mechanisms involved in the observed effects.

The molecular mechanism of activation cannot be elucidated in the complex cellular environment. The *in vitro* reconstitution of the GCN2-mediated ISR activation by the PERK inhibitors is the only

way to dissect the underlying mechanism. The finding that the PERK inhibitors increase the affinity of the GCN2 kinase for ATP is novel and important as it is likely relevant to other inhibitors and perhaps also to other kinases. In the revised manuscript, we show that not only PERK inhibitors but also PKR and 2 tyrosine-kinase inhibitors activate GCN2 by increasing its affinity for ATP.

The reviewer's interest in the cross-talks between the different domains of GCN2 (tRNA binding, kinase activation) is part of a bigger question, which is: what is the sequence of events between the different domains of GCN2 that lead to its activation. This is an important outstanding question. There have been decades of work on this theme and this has not been answered yet. One might also wonder whether the chemical activation of GCN2 and the natural activation of GCN2 occur via the same mechanism. Of note, there has been abundant literature in the recent years showing activation of GCN2 independently of tRNA binding.

How is GCN2 activated, what is the nature of the activating ligand (s) and what are the sequence of intramolecular events between the different domains of the protein leading to its activation are all very interesting questions but well beyond the scope of the present study.

3) Reviewer #1 It is not clear either what the source of purified eIF2 is.

Our response: We used a commercial recombinant eIF2 α from Sigma-Aldrich *EIF2S1* (#SRP5232). In the experiments for the revision, we have used our home-made recombinant eIF2 α and described its purification in the methods section.

4) Reviewer #1 Control experiments testing eIF2 phosphorylation in the absence of GCN2 should be included to exclude the possibility of phosphorylation by contamination of eIF2 with eIF2 kinases.

Our response: This has been tested. There is no phosphorylation of eIF2 α in absence of GCN2 in our assays. We have included this control as well as others, in a new supplementary figure (Supplementary Fig. S5).

5) Reviewer #1 It is of interest that ADP is more potent in activating GST-GCN2 than AMP-PNP even in the presence of 10x higher concentration of each compound than ATP (Suppl. Fig. S8). One would expect that the high concentration of each compound should result in full occupancy of

GCN2 and inhibition of its activity (at least for AMP-PNP). Thus, this Reviewer feels that the model requires thorough re-evaluation.

Our response:

The reviewer is correct in noting that in contrast to our work on the PERK inhibitors, which represents the main focus of the manuscript and for which we have provided a detailed mechanism, the mechanism of action of ADP and AMP-PNP in our in vitro assay had not been investigated in detail. The experiments presented in the former Fig. S8 were done as proof of concept to illustrate the notion that one should not expect all ATP derivatives to have the same GCN2 activating activity as the PERK inhibitors described in our manuscript. We didn't make any big claim: we just mention the findings as a clue collected on the way to dissecting the mechanism of action of the compounds we focus this manuscript on.

Following the reviewer's suggestion, we have now performed a full dose-response of GCN2 activation by ADP and AMP-PNP. ADP shows a similar Gaussian activation-inhibition curve as for the PERK inhibitors, whilst AMP-PNP does not activate the kinase but shows inhibition from 250 μ M. These additional findings further support our conclusions.

We thank the reviewer for the insightful comments that catalyzed the addition of controls and more in-depth explanations that led to a clearer and stronger version of our manuscript.

Reviewer #2 (Remarks to the Author)

The paper by Szaruga et al attempts to understand an interesting observation that the authors make that certain PERK inhibitors activated the integrated stress response (ISR). ISR signalling

is dependent upon 4 kinases that detect different signalling inputs -- this leads to eIF2a phosphorylation and subsequent brief protein translational attenuation. The two PERK inhibitors GSK157 and GSK 414 were seen to inhibit PERK but activate GCN2 in cells. In vitro experiments were used to further dissect the mechanism and showed that in the presence of ATP, GSK157 activated GCN2 to phosphorylate eIF2A and this followed a gaussian distribution. The authors put forth a mechanistic model based on this.

On the whole, the experiments are done well and that the data shows an interesting observation in cells. This work helps (at least partially) to understand observations where addition of inhibitors does not lead to downstream ISR signalling inhibition.

However, there are major points the authors need to address - The conclusions/claims of the paper particularly the model of activation is not supported by the data in my opinion and authors should either move this to the discussion along with adjusting their claims (this will affect the impact of the paper) or provide the necessary data.

Our response:

This reviewer raised important points that we will answer below. We would like to emphasize that the impact of this manuscript is broader than what is highlighted in the summary above. Our findings have transformative implications for our understanding of the ISR because most studies have been conducted at ISR-activating concentrations of these inhibitors. Hence, the efficacy of these molecules reported in various preclinical studies (~ 190) is likely to be due to ISR activation rather than the intended inhibition. This is a very important finding.

Although it is well known that ATP-competitive inhibitors of a given kinase often have off-target inhibitory activities towards other kinases, here we discover a distinct type of off-target activity: we find that an excess of inhibitor saturating the targeted kinase can result in activation of an off-target kinase. We discovered that this occurs because the ATP-competitive inhibitor directly activates the off-target kinase by increasing its affinity for ATP. To our knowledge, this represents a novel mechanism of kinase activation, of broad relevance because any kinase could be in principle activated in this way.

Because kinase inhibitors are broadly used as tool compounds to explore the biological function of kinases, the work presented here has broad relevance. The findings revealed in our manuscript call for an urgent systematic evaluation of **off-target kinase activating properties of existing kinase inhibitors**. This will not only help reduce the side effects of existing protein kinase inhibitors and refine their selectivity, but could also yield novel protein kinase activators.

We are grateful to the reviewer for requesting clarifications on the mechanism and additional information. We answered the reviewer's requests and feel that these additions add much clarity and strength to the manuscript.

1) Reviewer #2 The purified protein construct encompasses both the kinase domain and the pseudo-kinase domain. The interpretation of results is completely devoid of the mechanistic implications of pseudo-kinase domain. It is important to note that PERK does not have a pseudo-kinase domain (or any of the auto-inhibitory domains that GCN2 has). The authors do not demonstrate that the inhibitors bind to GCN2 ATP active site but assume this to be true based on its similarity to PERK. It could be that the inhibitors bind to pseudo-kinase site or another site on the protein and that this allosterically activates GCN2 kinase activity. Whilst the pseudo-kinase domain may not bind ATP due to the lack of Mg²⁺ coordination, it could bind the inhibitors. Allosteric regulation of kinases by the pseudo-kinase domain is well known although not completely understood. A previous study has shown GCN2 pseudo-kinase domain regulation of kinase activity (PMID 24811037) leads to enhanced eIF2 α activation. Thus, the authors should demonstrate if the inhibitors bind to the active site or the pseudo-kinase domain or both and include pseudo-kinase regulation of kinase activity in their assessment and interpretation.

Our response:

There are 2 components, a theoretical one with structure modeling and the experimental one: the activating compounds bind the kinase domain (KD) and do not bind the isolated pseudokinase domain (PKD).

The request of additional information from this reviewer is valuable. We realize that we didn't lay out all the evidence that backs up the interpretation of the data. The reviewer is also correct to note that we did not discuss the pseudokinase domain of GCN2 in the interpretation of the results. The choice of the recombinant protein used in the in vitro assay was based on work from others on GCN2 that have identified active fragments of the kinase (Ref 21-23). As explained below, the pseudokinase of GCN2 does not bind ATP and lacks residues involved in the binding of the PERK inhibitors. We are now presenting these additions in the revised manuscript.

Prior work has established that the pseudokinase domain (PKD) of GCN2 is inactive and does not bind ATP (Padyana, A. K., Qiu, H., Roll-Mecak, A., Hinnebusch, A. G. & Burley, S. K. Structural Basis for Autoinhibition and Mutational Activation of Eukaryotic Initiation Factor 2 α Protein Kinase GCN2. *J Biol Chem* **280**, 29289–29299 (2005); Murphy, J. M. *et al.* A robust methodology to subclassify pseudokinases based on their nucleotide-binding properties. *Biochem J* **457**, 323–334 (2013).) Murphy *et al.* demonstrated in 2014 that the mouse GCN2 pseudokinase domain does not bind ATP. The pseudokinase domain of GCN2 is highly conserved between mouse and human (Supplementary Fig. S8); both human and mouse GCN2 PKD lack nearly all the invariant kinase residues (Supplementary Fig. S8). Therefore, there is no need to re-evaluate whether the pseudokinase domain of GCN2 binds ATP.

It is reasonable to question whether the inhibitor could in principle bind to the pseudokinase

domain. Because of the lack of significant sequence similarity between PKD and KD of GCN2, we modelled the structure of the GCN2 PKD using AlphaFold2 and then superposed the structure of GCN2 KD (PDB 6n3n) and the GSK'157 compound docked on it with the AlphaFold2 model of PKD (Supplementary Fig. S8). Based on this superposition, we produced a structure-guided sequence alignment. This alignment clearly shows that the key residues involved in compound binding in the GCN2 KD differ in the PKD (Supplementary Fig. S8). The different amino acid composition of the GCN2 KD and PKD GSK'157 binding site (Supplementary Fig. S8) supports the interpretation that the PKD of GCN2 is not capable of binding this compound.

These comparisons between KD and PKD of GCN2 showing a lack of similarity are now added to the manuscript to explain why the idea that the molecules may bind to the PKD is improbable.

To experimentally test these structural analyses, we tested binding of GSK'157 to the recombinant GCN2 kinase or pseudokinase domains using thermal shift, a standard assay to assess compound binding. We found that GSK'157 shifted the melting temperature of the GCN2 kinase domain but not the pseudokinase domain (Fig. 4e and Supplementary Fig. S9). This reveals that GSK'157 binds the kinase domain of GCN2 but not the pseudokinase domain.

Similar findings were obtained with other activating compounds including new compounds added to the revised manuscript (Supplementary Fig. S9).

Importantly, we find that the mechanism uncovered with the broadly used PERK inhibitor GSK'157 also occurs with other, structurally diverse PERK inhibitors. Further, the same mechanism applies to the PKR inhibitor C16 and the unrelated tyrosine kinase inhibitors Dovitinib and Neratinib, exemplifying the broad relevance of our findings. Importantly, a crystal structure of Dovitinib bound to the kinase domain of GCN2 has been reported showing Dovitinib occupying the ATP-binding-pocket of GCN2, with an active-like conformation of the kinase⁴³. This fully supports the interpretation of our findings.

We acknowledge that the additional information provided above is important to support our interpretation. This is now explained in the revised results and discussion sections.

2) Reviewer #2 what is the affinity of GSK157 and GSK414 to GCN2? This is important information to understand what a sub-saturating concentration is?

Our response:

This is a very important omission on our part and we are grateful for this request that helps support the interpretation and the model.

The selectivity of GSK'157 was assessed at the time of its discovery by comparing its inhibitory activity towards various kinases. The reported inhibitory potencies toward PERK and GCN2 kinase domains were 0.9 nM and 3.162 μ M respectively (Atkins, C. *et al.* Characterization of a novel PERK kinase inhibitor with antitumor and antiangiogenic activity. *Cancer Research* **73**, 1993–2002 (2013). The potency of inhibition (IC₅₀) is driven by the binding affinity of the compound to its ATP-

binding pocket because it is a competitive inhibitor. Thus, the affinity of GSK'157 for GNC2 can be estimated to be $\sim 3 \mu\text{M}$. This suggests that at $\sim 3 \mu\text{M}$, GSK'157 occupies half the ligand binding sites on its target because the affinity (K_d or K_{obs} here) is defined by the concentration of a ligand at which half the ligand binding sites are occupied on the protein. The concentration of GSK'157 yielding maximum activation of GCN2 in our vitro assay was $\sim 5 \mu\text{M}$ (Fig 4c). At this concentration, \sim half of the ATP-binding sites of the dimeric GCN2 kinase are predicted to be occupied by the PERK inhibitors, yielding maximal activation. At lower concentrations, less than half of the ATP-binding sites of the GCN2 kinase are occupied and as a consequence, activation is below maximum. When the two ATP-binding sites are occupied by GSK'157, this results in kinase inhibition, because the activator outcompetes ATP, as we observed in (Fig 4b,c).

An alternative way to explain this is to start from the inhibition. Inhibition with ATP-competitive molecules is seen at full occupancy of the two ATP-binding pockets of the dimeric kinase. This occurs at saturating concentration of GSK'157. This thereby defines the saturating concentrations, revealed at full inhibition in Fig. 4c. Lowering the concentration (i.e., below saturation) yields activation. Thus, activation is attained at sub-saturating concentrations.

The same reasoning applies to AMG44, which has a reported potency to PERK of 6 nM and 3.900 nM to GCN2 (Smith, A. L. *et al.* Discovery of 1H-Pyrazol-3(2H) -ones as Potent and Selective Inhibitors of Protein Kinase R-like Endoplasmic Reticulum Kinase (PERK). *J Med Chem* **58**, 1426–1441 (2015). We observed maximum activation of GCN2 by AMG44 at $\sim 5 \mu\text{M}$ (Fig. 5). At this concentration (K_m), only one of the binding sites of the GCN2 kinase are predicted to be occupied with the PERK inhibitor.

We now reference prior knowledge on relative potencies of the inhibitors in the revised manuscript.

- 3) Reviewer #2 In the in vitro analysis the concentration of purified GCN2 is at 7.5nM, while the addition of ATP is at 6 μM . This is almost 1000 times excess of ATP. Similarly, the inhibitor concentration is micro molar range.

Our response:

The reactions are set for optimal kinetic studies with low (enzymatic) concentrations of enzyme, a large excess of substrate and at $\sim K_m$ for ATP. The PERK inhibitors are tested in a large concentration range to reveal their full properties.

- 4) Reviewer #2 The authors do not provide the affinity of the inhibitors to GCN2 but assuming that it maybe be close to that of PERK or even 10-fold less, this would still suggest all binding sites are saturated even at their lowest concentration of inhibitors.

Thus, the authors need to clarify why they suggest that this is sub saturating conditions?

Our response:

This is a good point. We have now clarified why activation is achieved at sub-saturating concentrations. We are more explicit in the revised manuscript. In the previous version, we mentioned that “their inhibitory activity towards PERK is at least 3000-fold more potent than for GCN2”. We agree with the reviewer that explaining the differences in affinities better will help the reader understanding the mechanism. We explained these details in the response to point 2) and 3) above.

As explained above, inhibition is seen at full occupancy of the ATP-binding pocket with ATP-competitive molecules, which defines the saturating concentration of GSK'157. Lowering the concentration (i.e., below saturation) yields activation. Thus, activation is attained at sub-saturating concentrations.

5) Reviewer #2 Following on from the above point – what is the affinity of ATP to GCN2?

Our response:

The K_m for ATP is $\sim 5 \mu\text{M}$ (Fig. 7b) which matches what others have reported.

6) Reviewer #2 If ATP is in excess this would out compete the non-specific PERK inhibitor? This makes it more important to state what the binding affinity is for the inhibitors to GCN2.

Our response:

Indeed: access of ATP overrides activation or in other words, the activity of activator is not observed with a fully active kinase. This is why the in vitro experiments are conducted at $\sim K_m$ for ATP.

To assess the idea of competition, we have tested if the PERK inhibitors could outcompete inhibition of GCN2 by the ATP-competitive GCN2 inhibitor A92. This turned out to be the case. These competition datasets are important as validation of the proposed mechanism. We thank the reviewer for inspiring this line of work.

Reviewer #2 In Fig4a – the ATP concentration is listed in mM but in fig 4b and fig4c the inhibitor concentrations are listed as μM . They should change the fig4a so that the ATP concentration is in μM same as fig 4b and fig 4c.

Our response:

This has been done.

7) Reviewer #2 In Fig 2d – there is no measurement of HRI protein levels.

Our response:

There are no commercially available HRI antibodies. Thus, we measured the efficacy of the HRI knockdown by qPCR and presented the results in Supplementary Fig. S2.

- 8) Reviewer #2 The amount of protein expressed in cells of PERK and GCN2 may differ thus small amounts of inhibitor may actually be saturating for PERK while sub-saturating for GCN2. The implications of this should be mentioned in the discussion.

Our response:

According to <http://mapofthecell.biochem.mpg.de/>, PERK is ~ 5.6 nM and GCN2 is ~ 28 nM in HeLa cells. This varies between different cell types as PERK abundance increases with the abundance of endoplasmic reticulum (secretory cells).

In light of the 3000-fold difference in the binding affinities of the PERK inhibitors to their primary target PERK and the off-target GCN2, the putative difference in abundance of the proteins becomes negligible. The relative potencies of the inhibitors for different targets have been clarified in the revised manuscript.

Reviewer #2 Other minor points:

- 1) The authors make a link between low concentration of inhibitor and that this corresponds to only one molecule in a dimer having inhibitor bound. What evidence do they have to state this? Why could not low levels of inhibitor bind to both protomers in a dimer of GCN2 and then not bind many other GCN2 dimers?

Our response:

The idea that low levels of inhibitor could bind both protomers of a dimer and not bind many others is theoretically possible but statistically improbable.

Moreover, if the inhibitor would preferentially bind both protomers in a dimer and not bind many other GCN2 dimers, a dose response of the inhibitor would only result in inhibition because binding of ATP or the inhibitor to the activate site is mutually exclusive.

- 2) I am not entirely sure what the point of the gaussian curve is as it will change if you increase any of the limiting factors i.e., if you add more ATP then the curve will shift?

Our response:

The Gaussian activation-inhibition curve shows activation followed by inhibition, an important aspect of the mechanism. ATP is essential for kinase activation. When all ATP-binding sites of the kinase are occupied by the ATP-competitive molecule GSK'157, the binding of ATP is prevented and the kinase is inhibited. This occurs when the inhibitor saturates the binding sites on the kinase (saturating concentrations). Lowering the concentration (i.e., below saturation) yields activation. Thus, activation is attained at sub-saturating concentrations. Thus, the Gaussian curve reveals the mechanism of activation and sub-saturating concentration. This is explained better in the revised manuscript.

To strengthen the mechanism, we have now conducted competition experiments. We show that the GCN2 activating compounds outcompete inhibition by the GCN2 ATP-competitive inhibitor A92. For this inhibition to be alleviated, the inhibitory compound has to be removed from the ATP-binding pocket. We feel that these experiments are a valuable addition that strengthen the interpretation of our findings.

3) In certain figures the difference between activated and inactivated bands are very difficult to interpret. Why not run the gel for longer and thus allow greater separation between active and inactive bands?

Our response:

We do not want to change the experimental design here, which has been optimized and standardized for the purpose of this study.

As the reviewer might suspect, it is not trivial to separate the active and inactive GCN2. The resolution in the figure presented is fit for the purpose, i.e., the bands are distinct and can be quantified. If we run the gels longer, the $\sim 37\text{kDa}$ eIF2 α substrate will run out of the gel, as it currently runs close to the migration front.

We thank this reviewer for his/her insightful review. We answered the reviewer's queries and feel that the additional experiments generated a clearer and stronger study.

Reviewer #3 (Remarks to the Author):

The Integrated Stress Response (ISR) is a central homeostatic network that maintains cellular health. Over the last few years, dysregulation of the ISR has been associated with neurodegeneration and cognitive disorders. Therefore, the ISR emerges as an attractive target for pharmacological intervention in human disease. In this study, Szaruga et al., reported that commercially available small molecules designed to specifically inhibit the ISR sensor PERK activate the ISR kinase GCN2 at high concentrations. By combining in vivo studies with biochemical assays, the authors showed that the GSK's PERK inhibitors stimulate GCN2 activity both in vitro and in vivo. Similarly, the GCN2 inhibitor A92 activates PERK at a high dosage.

Overall, the manuscript is well-written, the experiments are high quality. However, some of the authors' claims are not fully supported by the data and additional experiments are required to support their conclusions. Here below there are a few points that should be addressed.

My major concern is that the evidence provided to support the model of activation of GCN2 by GSK'157 is weak. Several other possibilities could explain the authors' conclusions.

Our response: This reviewer raised important points that required clarification. We have addressed all his/her concerns in the response below and in the revised manuscript.

1) Reviewer #3 The authors should test kinase extrinsic and intrinsic mechanisms. Is there an inhibitor-induced feedback loop? To test this possibility, the authors should test the effect of the inhibitors in cells in which eIF2 cannot be phosphorylated. As per kinase intrinsic mechanism, the authors should test for changes in the kinase itself, including the use of catalytically dead mutant PERK or mutants in which the inhibitor cannot bind (Shokat method).

Our response: The answers to the reviewer's questions above are found in the existing experiments. For better clarity, we have reworded the presentation of the results and their discussion.

The GSK inhibitors activate ISR and GCN2 by a kinase **intrinsic** mechanism. We show that **the knockdown of GCN2 completely abrogated ISR activation by the PERK inhibitors demonstrating unambiguously that the activities measured are entirely mediated by GCN2.** Moreover, we show that phosphorylation of the substrate eIF2 α is lost in cells where the expression of GCN2 has been knocked-down. Thus, there is no room for an alternative mechanism: the reported activating activities of the compounds are entirely mediated by a GCN2-intrinsic mechanism.

Having established the above, conducting experiments in cells in which eIF2 \$\alpha\$ cannot be phosphorylated will not be informative in this context. eIF2 \$\alpha\$ is the target. We find more interesting

to dissect the upstream event because it directly focuses on the mechanism rather than its consequence.

Moreover, we have shown that the PERK inhibitors activate GCN2 in cells where expression of PERK has been knockdown. This demonstrates that the activity of the inhibitors we report here is independent of PERK. Therefore, there is no need to test PERK mutants in this study where PERK is not involved.

2) Reviewer #3 Does high concentrations of GSK'157 increase ER membrane localization of PERK?

Our response: PERK is an ER resident transmembrane protein. Its localization does not change. Moreover, we show that the GCN2-activating activity of the PERK inhibitors is independent of PERK. Therefore, the localization of PERK is not relevant to our study.

Question 1 & 2 from this reviewer made us realize that we didn't emphasize enough that the activity of the inhibitors was independent of PERK. This is now corrected in the revised manuscript.

3) Reviewer #3 Kinetic analysis shows that GSK'157 lowers the threshold of activation of GCN2. The authors interpreted this finding as reasoning that GSK'157 is an allosteric GCN2 activator. They propose that GSK binds to GCN2 protomer in the ATP pocket favoring ATP binding to the other protomer. However, GCN2 forms constitutive dimers in an inactive antiparallel conformation. Activation of the kinase requires a conformational change that re-organizes GCN2 dimers into a parallel active state. Thus, a possible scenario is that GSK compounds stabilize GCN2 in a parallel conformation.

Our response: The mechanism of activation is not solely driven by the kinetic analysis but a large dataset. Yes, it is possible that as a consequence of the activation mechanism we describe here, the kinase is stabilized in a parallel active state. Our findings revealed that the affinity of GCN2 for ATP is increased in the presence of activators. We propose that binding of ATP to one protomer transduces a signal to the other protomer resulting in increased affinity for ATP. This interpretation builds up on the structure of GCN2 from Hinnebusch and Burley, which has revealed that the active kinase has an opened ATP-binding cleft that increases ATP binding. We indeed think that this is a consequence of an activation event triggered by the compound. We now have added a sentence in the discussion to mention the possibility that activation could involve a switch from the antiparallel to the parallel conformation.

Importantly, we have demonstrated that the kinase is a dimer and addition of the compound does not change its dimerization status (Fig. 7a and Supplementary Fig. S4 and S10). We have further highlighted this finding in the revised manuscript because one initial hypothesis accounting for the activation mechanism was oligomerization. The data shows it is not the case. The dimeric status

of the kinase is now presented in the main figure to ensure that readers will not miss this important information. We thank the reviewer for prompting this change.

4) Reviewer #3 Additionally, the ISR sensor kinases tend to form oligomers to regulate their activity. Another non-mutual exclusive possibility is that GSK'157 favors GCN2 high-order assembly increasing the activity.

Our response: as stated above, our thinking was aligned with the reviewer and the idea that the inhibitors may increase oligomerization of the kinase was the first hypothesis we explored. We have demonstrated that the kinase is a dimer and addition of the compound does not change its dimerization status (Fig. 7a and Supplementary Fig. S4 and S10). There is more emphasis on this point in the revised manuscript with the dimeric status of the kinase now presented in the main figure to ensure that readers will not miss this important information. After we found that the compound does not change GCN2 oligomerization, we next searched for an alternative explanation and discovered that GSK'157 increased affinity for ATP.

5) Reviewer #3 Finally, GCN2 contains a pseudokinase domain (YKD) that is required to stabilize and boost kinase activity. The recombinant protein used in this study for the in vitro kinase assay lacks the t-RNA binding domain but maintains the YKD. In this scenario, GSK'157 might bind to the YKD stabilizing KD-YKD interactions.

The authors should do additional experiment to rule in or out the different scenarios described above.

There are 2 components, a theoretical one with structure modeling and the experimental one: the activating compounds bind the kinase domain (KD) and do not bind the isolated pseudokinase domain (PKD).

The request of additional information from this reviewer is valuable. We realize that we didn't lay out all the evidence that backs up the interpretation of the data. The reviewer is also correct to note that we did not discuss the pseudokinase domain of GCN2 in the interpretation of the results. The choice of the recombinant protein used in the in vitro assay was based on work from others on GCN2 that have identified active fragments of the kinase (Ref 21-23). As explained below, the pseudokinase of GCN2 does not bind ATP and lacks residues involved in the binding of the PERK inhibitors. We are now presenting these additions in the revised manuscript.

Prior work has established that the pseudokinase domain (PDK) of GCN2 is inactive and does not bind ATP (Padyana, A. K., Qiu, H., Roll-Mecak, A., Hinnebusch, A. G. & Burley, S. K. Structural Basis for Autoinhibition and Mutational Activation of Eukaryotic Initiation Factor 2 α Protein Kinase GCN2. *J Biol Chem* **280**, 29289–29299 (2005); Murphy, J. M. *et al.* A robust methodology to subclassify pseudokinases based on their nucleotide-binding properties. *Biochem J* **457**, 323–334

(2013).) Murphy et. al demonstrated in 2014 that the mouse GCN2 pseudokinase domain does not bind ATP. The pseudokinase domain of GCN2 is highly conserved between mouse and human (Supplementary Fig. S8); both human and mouse GCN2 PKD lack nearly all the invariant kinase residues (Supplementary Fig. S8). Therefore, there is no need to re-evaluate whether the pseudokinase domain of GCN2 binds ATP.

It is reasonable to question whether the inhibitor could in principle bind to the pseudokinase domain. Because of the lack of significant sequence similarity between PKD and KD of GCN2, we modelled the structure of the GCN2 PKD using Alphafold2 and then superposed-the structure of GCN2 KD (PDB 6n3n) and the GSK'157 compound docked on it with the Alphafold2 model of PKD (Supplementary Fig. S8). Based on this superposition, we produced a structure-guided sequence alignment. This alignment clearly shows that the key residues involved in compound binding in the GCN2 KD differ in the PKD (Supplementary Fig. S8). The different amino acid composition of the GCN2 KD and PKD GSK'157 binding site (Supplementary Fig. S8) supports the interpretation that the PKD of GCN2 is not capable of binding this compound.

These comparisons between KD and PKD of GCN2 showing a lack of similarity are now added to the manuscript to explain why the idea that the molecules may bind to the PKD is improbable.

To experimentally test these structural analyses, we tested binding of GSK'157 to the recombinant GCN2 kinase or pseudokinase domains using thermal shift, a standard assay to assess compound binding. We found that GSK'157 shifted the melting temperature of the GCN2 kinase domain but not the pseudokinase domain (Fig. 4e and Supplementary Fig. S9). This reveals that GSK'157 binds the kinase domain of GCN2 but not the pseudokinase domain.

Similar findings were obtained with other activating compounds including new compounds added to the revised manuscript (Supplementary Fig. S9).

Importantly, we find that the mechanism uncovered with the broadly used PERK inhibitor GSK'157 also occurs with other, structurally diverse PERK inhibitors. Further, the same mechanism applies to the PKR inhibitor C16 and the unrelated tyrosine kinase inhibitors Dovitinib and Neratinib, exemplifying the broad relevance of our findings. Importantly, a crystal structure of Dovitinib bound to the kinase domain of GCN2 has been reported showing Dovitinib occupying the ATP-binding-pocket of GCN2, with an active-like conformation of the kinase⁴³. This fully support the interpretation of our findings.

We acknowledge that the additional information provided above is important to support our interpretation. This is now explained in the revised results and discussion sections.

6) Reviewer #3 Treatment with the GCN2 inhibitor A92 seems to induce PERK at a high dosage. Does A92 activate the UPR? PERK and PKR have a high degree of homology.

Our response: We show that high concentration of A92 activates PERK as well as ISR signalling i.e., eIF2 in cells (Fig. 3g-i). Furthermore, we show that PERK activation is abrogated when PERK

is silenced, revealing that the activation of PERK by A92 is entirely mediated by PERK (Supplementary Fig. S3).

7) Reviewer #3 To claim generality of the process, the authors should test at least an additional ISR inhibitor (C16) and see whether the same behavior takes place. Moreover, does A92 show activity towards PKR? Measuring the effect of A92 on the other ISR kinases would strengthen the findings.

Our response: This is an excellent suggestion. We have now tested C16 in a range of assays and added these new datasets to the revised manuscript. The result section on C16 is pasted below for convenience.

To further explore the generalizable potential of our findings to diverse ISR kinase inhibitors, we tested if the PKR inhibitor C16²⁹ could activate GCN2 in vitro. We found that C16 activated GCN2 in vitro between 0.19 and 1.5 μ M and inhibited the kinase at high concentrations (Fig. 6a,b). Thus, the PKR inhibitor activated GCN2, following a Gaussian activation-inhibition curve in vitro. C16 also activated GCN2 and the ISR in cells up to 6 μ M (Fig. 6b). At 12 μ M, ISR induction by C16 appeared reduced, relative to 6 μ M (Fig. 6b). Higher concentrations could not be tested in a meaningful way in cells because they resulted in precipitation of the compound in the cell culture media. Importantly, thermal shift experiments revealed that C16, like the PERK inhibitors, bound the kinase domain of GCN2 but not its pseudokinase domain (Supplementary Fig. S9a,b). To gain insights into the mechanism, we tested if C16 could outcompete GCN2 inhibition by the ATP-competitive inhibitor A92, as previously observed for PERK inhibitors. We found that 1.25-40 μ M C16 outcompeted GCN2 inhibition by A92, whilst inhibiting GCN2 at concentrations greater than 80 μ M (Fig. 6c). This shows that not only diverse PERK inhibitors but also a PKR inhibitor can activate GCN2 and the ISR through a similar concentration-dependent mechanism.

We thank the reviewer for his/her suggestions that contributed to generate new figures.

Reviewer #3: Other points

1) The in vivo experiment convincingly shows that the administration of tunicamycin in the presence of the PERK inhibitor GSK'157, blocks PERK activity (i.e., autophosphorylation) but does not have any effect on the phosphorylation of the PERK substrate eIF2 α . To gain insight into these conflicting results, the authors used cell culture to test the efficacy of GSK'157 in blunting ISR signalling upon tunicamycin treatment. Dose-response analysis revealed that GSK'157 activates the ISR at a high dose. To justify their focus on GCN2, in lines 23-30 of page 3, the authors state "Searching for the molecular basis of this phenomenon, we reasoned that PERK inhibition might overwhelm another eIF2 α kinase. Accumulation of misfolded proteins during unresolved ER stress

could prevent their degradation and thereby diminish the pool of amino acids, a signal to activate GCN2, the amino-acid sensing ISR kinase^{17,18}. We found that, in Tm-stressed cells, ISR-activating concentrations of GSK'157 caused robust activation of GCN2 detected by its phosphorylation and reduced mobility on SDS-PAGE (Fig. 1f). This demonstrates that, in the presence of ER stress, the benchmark PERK inhibitor GSK'157 activates GCN2 and the ISR in the low micromolar range.”

Tunicamycin is a pleomorphic drug that activates PKR (PMID: 19007793). To my knowledge, there is no formal evidence that the inhibition of PERK upon UPR induction is compensated by GCN2 and the experiments showed in figure 1 do not rule out this possibility. Therefore, the conclusion of the first paragraph should be rephrased.

Our response: The reviewer is correct, there is no evidence to support the hypothesis we put forward regarding a possible indirect GCN2 activation upon PERK inhibition. It was a hypothesis that we examined and led to the discovery that GCN2 was activated. Thus, we ruled out that GCN2 activation occurred by the indirect mechanism we initially proposed when we subsequently discovered that PERK inhibitors directly activate GCN2 independently of PERK in cells (Fig. 2) and in vitro (Fig. 4).

We want to keep this narrative. It explains the thought process that went into looking at GCN2 activation.

The conclusion “This demonstrates that, in the presence of ER stress, the benchmark PERK inhibitor GSK'157 activates GCN2 and the ISR in the low micromolar range.” is correct and doesn't need to be changed.

Moreover, the manuscript would benefit by showing the activity (i.e., autophosphorylation) of all ISR kinases in cells co-treated with tunicamycin and increasing concentration of GSK'157.

Our response: We have examined the involvement of each ISR kinases genetically rather than biochemically to unambiguously demonstrate which are central and which are not involved in the mechanisms reported.

Assessing kinase activation by measuring autophosphorylation can generate false negative results because maximal ISR activation is often observed at substoichiometric autophosphorylation of ISR kinases. We have highlighted this caveat in a Method review we wrote on ISR detection (Krzyszosiak, A., Pitera, A. P. & Bertolotti, A. The Integrated Stress Response, Methods and Protocols. *Methods Mol Biology* **2428**, 3–18 (2022)). Therefore, we turned to unambiguous genetic assessment of the kinase involvement to generate conclusive results. We have now highlighted this better in the revised manuscript.

2)Figure 2C. The blot shows high levels of p-eIF2 α at a steady state (line 1). Has the control been treated with a cationic lipid transfectant reagent? One possibility might be that the

transfection/handling activated the ISR. While the p-eIF2 α WB might lead to misinterpretation, ATF4 expression results are very clear. The authors should consider replacing the panel.

Our response: Yes, the control has been treated with a transfectant reagent to ensure that the differences between the distinct conditions come solely from siRNA treatment and not from different experimental set ups.

The dynamic range of phosphorylation of eIF2 α is small. Therefore, we don't use P-eIF2 α immunoblots as sole evidence for ISR changes. Note that the immunoblot images are provided without manipulation of contrast. We have added this note in our figure legends to explain to the readers why our immunoblots show differences that may be interpreted as more subtle than in other published papers. The differences are real. We have to explain this point very often and thus have dedicated an entire review to the difficulties associated with ISR detection (Kryzosiak, A., Pitera, A. P. & Bertolotti, A. The Integrated Stress Response, Methods and Protocols. *Methods Mol Biology* **2428**, 3–18 (2022)). As with any signalling pathway, the signal is amplified at each step. Thus, minute changes in eIF2 α phosphorylation result in robust biological changes. Yet, by nature, the primary signal in this signalling cascade, eIF2 phosphorylation, is difficult to measure quantitatively. Thus, we use several ISR markers to assess ISR induction. The reviewer is correct in highlighting that ATF4 changes provide a more obvious readout. This is generally the case.

3) Figure 2D. p-eIF2 α levels in the blot relative to PERK KD are hard to see due to a technical artefact. Please replace the image.

Our response: This has been replaced.

4) Figure S2. Validation of KD efficiency for GCN2, PKR, and PERK is missing.

Our response: The KD efficiency of GCN2, PERK and PKR has been validated and shown by immunoblot Fig. 2d and by qPCR for HRI Supplementary Fig. S2 because there are no commercially available HRI antibodies.

5) Figure 3G. The experiment shown in figure 3G indicates that the GCN2 inhibitor A92 activated PERK at a high dosage. The activity of A92 on GCN2 was assayed in cells in steady-state conditions since p-GCN2 was already visible in the absence of stress. The authors reasoned that the p-GCN2 observed is due to kinase basal activity. If this is true, then why p-GCN2 basal levels are undetectable in Figures 3A and

Our response: As explained above, ISR activation often occurs with substoichiometric activation of an ISR kinase. Careful examination of the Fig 3G, at basal levels, shows basal p-GCN2 and a band of low intensity of slower mobility than the main GCN2 in the GCN2 panel: this band is the active GCN2.

The basal levels of p-GCN2 are detectable but of lower intensity in Fig., 3A, pasted below for reference.

Thus, our interpretation stands.

We thank this reviewer for inspiring new experiments that generated new figures in the manuscript and for his/her request of clarifications.

REVIEWERS' COMMENTS

Reviewer #1 (Remarks to the Author):

The revised version of the manuscript has been improved. However, the important question remains about the physiological relevance of the findings to help readers. Can the authors show that combined treatments with PERKi and GCN2i compromises the survival of ER stressed cells to the same level as cells deficient for p-eIF2? Such an experiment is largely feasible with p-eIF2 proficient and deficient MEFs. The result would be of great help to scientists in the ISR field.

Reviewer #2 (Remarks to the Author):

This reviewer is satisfied with the authors amendments that now strengthens the manuscript.

Reviewer: The revised version of the manuscript has been improved. However, the important question remains about the physiological relevance of the findings to help readers. Can the authors show that combined treatments with PERKi and GCN2i compromises the survival of ER stressed cells to the same level as cells deficient for p-eIF2? Such an experiment is largely feasible with p-eIF2 proficient and deficient MEFs. The result would be of great help to scientists in the ISR field.

Our response: This is a completely new request which wasn't part of the first review. The physiological relevance of our findings is demonstrated in Figure 1, 2 and 3. In addition, ~ 200 papers have used ISR kinase inhibitors at the concentrations we describe here to be ISR-activating. Thus, the physiological relevance of our findings is backed up by a large body of evidence.

Assessing cell viability of ER stressed cells in the presence of ISR kinase inhibitors is a completely new question from this reviewer and beyond the scope of the paper. Whilst the initial experiments were done in the presence of an ER stressor, the rest of the paper is stress independent. Indeed, we show that the ISR-activating properties of the compounds are stress independent and discover the underlying mechanism. We have a lot of experience on quantitative assessment of cell survival of ER stress cells in presence of absence of compounds (1. Tsaytler, P., Harding, H. P., Ron, D. & Bertolotti, A. Selective inhibition of a regulatory subunit of protein phosphatase 1 restores proteostasis. *Science* **332**, 91–94 (2011), 2. Das, I. *et al.* Preventing proteostasis diseases by selective inhibition of a phosphatase regulatory subunit. *Science* **348**, 239–242 (2015). 3. Luh, L. M. & Bertolotti, A. Potential benefit of manipulating protein quality control systems in neurodegenerative diseases. *Current Opinion in Neurobiology* **61**, 125–132 (2020)). Whilst this new experiment suggested by this reviewer is indeed feasible, we believe it will not be informative. This is because the kinase inhibitors have other off-target activities that will render the experiment inconclusive.

We followed the editor's recommendation and have edited the discussion of the manuscript to highlight the physiological relevance of the findings.